# Robust Preference Alignment via Directional Neighborhood Consensus

**Ruochen Mao**[1*] **Yuling Shi**[2] **Xiaodong Gu**[2] **Jiaheng Wei**[1,3†]
[1]The Hong Kong University of Science and Technology (Guangzhou)
[2]Shanghai Jiao Tong University [3]D5 Data
ruochenmao@163.com, yuling.shi@sjtu.edu.cn,
xiaodong.gu@sjtu.edu.cn, jiahengwei@hkust-gz.edu.cn

## ABSTRACT

Aligning large language models with human preferences is critical for creating reliable and controllable AI systems. A human preference can be visualized as a high-dimensional vector where different directions represent trade-offs between desired attributes (e.g., helpfulness vs. verbosity). Yet, because the training data often reflects dominant, average preferences, LLMs tend to perform well on common requests but falls short in specific, individual needs. This mismatch creates a *preference coverage gap*. Existing methods often address this through costly retraining, which may not be generalized to the full spectrum of diverse preferences. This brittleness means that when a user's request reflects a nuanced preference deviating from the training data's central tendency, model performance can degrade unpredictably. To address this challenge, we introduce Robust Preference Selection (RPS), a post-hoc, training-free method by leveraging *directional neighborhood consensus*. Instead of forcing a model to generate a response from a single, highly specific preference, RPS samples multiple responses from a local neighborhood of related preferences to create a superior candidate pool. It then selects the response that best aligns with the user's original intent. We provide a theoretical framework showing that, under mild conditions where (i) nearby preference directions correspond to better-trained regions of the model and (ii) the reward-model scores change smoothly with small angular changes in the preference vector, our neighborhood generation strategy yields a higher expected best score than a strong baseline that also samples multiple candidates. Comprehensive experiments across three distinct alignment paradigms (DPA, DPO, and SFT) demonstrate that RPS consistently improves robustness against this baseline, achieving win rates of up to 69% on challenging preferences from under-represented regions of the space without any model retraining. Our work presents a practical, theoretically-grounded solution for enhancing the reliability of preference-aligned models.

## 1 INTRODUCTION

Aligning large language models (LLMs) with human preferences is crucial for creating reliable and controllable AI systems (Ouyang et al., 2022; Christiano et al., 2017; Ziegler et al., 2020; Zhu et al., 2023). User preferences can be modeled in a multi-dimensional space where different directions represent trade-offs between desired attributes, such as helpfulness versus verbosity (Wang et al., 2024a; Dong et al., 2023). As illustrated in Figure 1, this creates a foundational challenge: the **preference coverage gap**. While the space of potential user preferences is vast and diverse, as depicted in Figure 1(a), the alignment process often optimizes for a dominant, average preference, meaning the training data is concentrated in a narrow region (Figure 1(b)). This focus on average preferences makes models brittle; when faced with a user preference that reflects more individual needs and deviates from this central tendency—a common out-of-distribution (OOD) challenge—their performance can degrade unpredictably, undermining user trust (Hendrycks et al., 2020).

---

*Work done during a research internship at HKUST (GZ).
†Corresponding author.

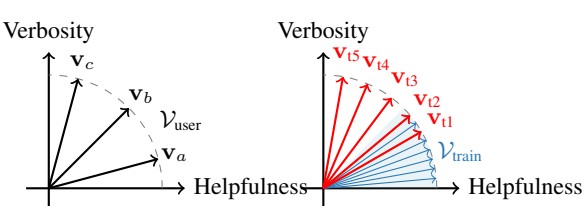

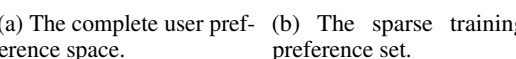

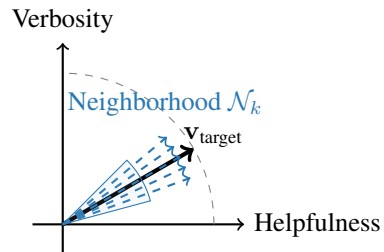

(a) The complete user preference space.

(b) The sparse training preference set.

Figure 1: Illustration of the preference coverage gap. While user preferences (a) can span the entire space, the model's training data (b) is often concentrated on dominant, average preferences, leaving individual needs in a sparse subset. This creates a gap where a user's target preference may lie.

Figure 2: Conceptual visualization of RPS. Instead of relying on a single, potentially out-of-distribution target preference $\mathbf{v}_{\text{target}}$ (solid black arrow), RPS samples $k$ directions from its local neighborhood (dashed blue arrows). By generating responses from this diverse set, RPS can identify a response that better aligns with the user's true intent.

To address this coverage gap, many existing solutions focus on training-time interventions. These include methods like data augmentation or the adoption of principles from Distributionally Robust Optimization (DRO) (Duchi et al., 2021; Ben-Tal et al., 2013; Duchi & Namkoong, 2021) to create models that are resilient to shifts in preference distributions (Xu et al., 2025). While effective, such approaches often require costly retraining cycles and may still fail to generalize to the full spectrum of diverse, individual preferences. This motivates an alternative question: can we enhance robustness at **inference time**, without any modification to the underlying model?

This paper argues that forcing a model to generate a response from a single, highly specific and less common preference direction is inherently fragile. We propose a paradigm shift from direct generation to one based on **directional neighborhood consensus**. As visualized in Figure 2, instead of attempting to extrapolate to a specific, under-represented preference point, it is more robust to explore the local neighborhood, generate responses from these more dominant, better-understood directions, and then select the one that best satisfies the original preference.

To realize this, we introduce **Robust Preference Selection (RPS)**, a post-hoc adjustment method that enhances preference alignment at inference time without any retraining. RPS first samples a set of candidate preference vectors from the neighborhood of the user's target preference. It then generates a response for each of these nearby vectors and, finally, uses the target preference itself as a criterion to select the optimal response from this diverse set. This approach effectively leverages the model's existing capabilities in well-trained regions of the preference space to satisfy requests in undertrained ones[1].

Our contributions are threefold:

- We formally define the *preference coverage gap* as a critical out-of-distribution (OOD) challenge that undermines the reliability of aligned LLMs. To address this, we introduce RPS, a novel, training-free method that enhances robustness through post-hoc adjustment without requiring any model modification.

- We propose RPS, a method grounded in neighborhood consensus, and provide a theoretical framework proving that its neighborhood generation strategy is superior to a strong multi-candidate baseline.

- We conduct extensive experiments across three distinct alignment paradigms (DPA, DPO, and SFT) and three datasets (`UltraFeedback`, `HelpSteer`, and `HelpSteer2`). Our results show that RPS consistently improves robustness, achieving win rates of up to 69% on challenging OOD preferences and demonstrating its broad applicability.

---

[1]Code and data available at `https://github.com/rcmao/robust-preference-alignment`.

## 2 RELATED WORK

### 2.1 PREFERENCE ALIGNMENT IN LARGE LANGUAGE MODELS

Aligning LLM behavior with human preferences has become a central research area. Reinforcement Learning from Human Feedback (RLHF) is a pioneering pipeline that fine-tunes models with human preference rankings, as demonstrated by (Ouyang et al., 2022). However, RLHF compresses diverse user preferences into a single scalar reward and requires complex reward modeling plus reinforcement learning. To simplify this process, Direct Preference Optimization (DPO) was introduced (Rafailov et al., 2023), which recasts preference optimization as supervised classification, eliminating the need for explicit reward models. Subsequent generalizations explore divergence families and latent user heterogeneity (Yang et al., 2023a; Chidambaram et al., 2024), while others have proposed new theoretical paradigms for understanding preference learning (Azar et al., 2024). Moving beyond scalar objectives, Directional Preference Alignment (DPA) enables users to specify trade-offs in a multi-axis reward space (Wang et al., 2024a). Similarly, SteerLM conditions supervised fine-tuning on attribute labels, exposing controllable style dimensions such as helpfulness or humor (Dong et al., 2023). These methods are part of a broader research effort in controllable text generation, which aims to provide users with fine-grained control over model outputs (Liang et al., 2024). Our work differs from these training-time approaches: rather than modifying the model weights, we focus on inference-time robustness to preference shifts through directional neighborhood consensus.

### 2.2 ENHANCING ROBUSTNESS IN LANGUAGE MODELS

While the alignment methods described above are powerful, a key challenge remains: models often remain brittle under out-of-distribution (OOD) preferences. Recent work has formalized preference distribution shifts and proposed distributionally robust objectives such as (Xu et al., 2025), which strengthen resilience during training. Beyond alignment, the broader NLP community has highlighted the challenges of OOD generalization, with benchmarks such as (Yang et al., 2023c;b). At inference time, an alternative approach is to use ensemble-like methods, a principle with deep roots in machine learning (Dietterich, 2000). For instance, (Wang et al., 2022) shows that sampling diverse reasoning paths and aggregating their consensus yields more reliable results. The principle of post-hoc adjustment for robustness is also explored in other domains, such as classification, where scaling model outputs can mitigate the effects of distributional shifts (Wei et al., 2023).

Extending this idea, recent inference-time alignment frameworks share our post-hoc perspective but differ in mechanism. Many rely on direct intervention in the generation process through token-level guidance or activation steering (Li et al., 2025; Shahriar et al., 2024), or require auxiliary models for decoding-time guidance (Chehade et al., 2025; Chandra et al., 2025). In contrast, our RPS approach operates purely in the preference space. By leveraging neighborhood consensus to select an optimal response, it avoids direct manipulation of the model's internal states, offering a simpler and more black-box solution that requires no external guidance models.

## 3 PROBLEM SETUP AND THEORETICAL FRAMEWORK

We build upon the problem formulation of Directional Preference Alignment (DPA) (Wang et al., 2024a). In this section, we formalize the preference alignment challenge by first defining the preference space and characterizing the coverage gap that causes model brittleness. We then establish the theoretical foundations for our proposed solution, Robust Preference Selection (RPS).

### 3.1 PREFERENCE SPACE AND REWARD MODEL

We model user preferences in a two-dimensional space for clarity of illustration, as depicted in Figure 1(a), spanned by two key axes: **Helpfulness** and **Verbosity** (Wang et al., 2024a; Dong et al., 2023). While our experiments focus on this low-dimensional (2D) setting that is standard in current alignment work, the theoretical framework is stated for a general $d$-dimensional preference vector $\mathbf{v} \in \mathbb{S}^{d-1}$. To quantify these attributes, we formalize the notion of a reward vector.

**Definition (Reward Vector):** A reward model maps a prompt-response pair $(x, y)$ to a reward vector $\mathbf{r}(x, y) = (r_h(x, y), r_v(x, y)) \in \mathbb{R}^2$. The components $r_h(x, y)$ and $r_v(x, y)$ are scalar scores representing the helpfulness and verbosity of the response, respectively (Wang et al., 2024a).

For all experiments, we use the publicly available `RewardModel-Mistral-7B-for-DPA-v1`[2]; further details on the scoring procedure are provided in Appendix A.9.

A user's preference is represented as a normalized direction vector $\mathbf{v} = (v_h, v_v) \in \mathbb{S}^1$ on the unit circle, where $v_h$ and $v_v$ specify the desired weights for helpfulness and verbosity. This can be parameterized by an angle $\theta$, such that $\mathbf{v} = (\cos\theta, \sin\theta)$. The goal of a preference-aligned model is to generate a response $y$ that maximizes the projected reward: $\mathbf{v}^T\mathbf{r}(x, y)$. Our framework assumes that this reward model $\mathbf{r}(x, y)$ is well-calibrated and provides meaningful scores across the entire preference space, including for out-of-distribution directions.

## 3.2 THE PREFERENCE COVERAGE PROBLEM

The central challenge in preference alignment, as illustrated in Figure 1, is the discrepancy between the vast space of user preferences and the limited coverage of the training data. We formalize this problem as follows:

**Definition 1 (User Preference Space).** Let $\mathcal{V}_{\text{user}}$ denote the complete set of all possible normalized preference vectors $\mathbf{v} \in \mathbb{S}^1$. This represents the entire spectrum of potential user preferences, as depicted in Figure 1(a).

**Definition 2 (Training Preference Set).** Let $\mathcal{V}_{\text{train}} \subset \mathcal{V}_{\text{user}}$ be the subset of preference directions used during training, visualized as the concentrated region in Figure 1(b). This set is often sampled from a constrained range.

**Definition 3 (Preference Coverage Gap).** The coverage gap, illustrated by the difference between the full space in Figure 1(a) and the training data in Figure 1(b), consists of all preference vectors that are not within an $\epsilon$-neighborhood of any training vector: $\text{Gap} = \mathcal{V}_{\text{user}} \setminus \mathcal{N}_\epsilon(\mathcal{V}_{\text{train}})$.

When a target preference $\mathbf{v}_{\text{target}}$ lies in this gap—as illustrated with the out-of-distribution vector in Figure 1(b)—the model's performance is unreliable. Our goal is to develop a method that can robustly generate a high-quality response $y^*$ that maximizes user satisfaction, even for out-of-distribution preferences:

$$y^* = \arg\max_y \mathbf{v}_{\text{target}}^T\mathbf{r}(x, y), \tag{1}$$

where $\mathbf{r}(x, y) = (r_h(x, y), r_v(x, y))$ represents the helpfulness and verbosity scores of response $y$ to prompt $x$. This challenge of performing well on a target preference $\mathbf{v}_{\text{target}}$ that lies in the gap can be framed through the lens of Distributionally Robust Optimization (DRO) (Duchi et al., 2021; Ben-Tal et al., 2013; Duchi & Namkoong, 2021). In the DRO paradigm, the objective is to find a policy that is robust not just to the empirical training distribution (represented by $\mathcal{V}_{\text{train}}$) but to a family of plausible test distributions. Our inference-time approach complements training-time DRO solutions by addressing this distributional shift post-hoc, at the point of generation.

## 3.3 NEIGHBORHOOD CONSENSUS THEORY

Instead of merely justifying the final selection step, our theoretical framework aims to explain why the entire neighborhood generation strategy is superior to a strong baseline that repeatedly samples from the target direction. The core intuition is that for an out-of-distribution (OOD) preference $\mathbf{v}_{\text{target}}$, the model's performance is degraded. By sampling from a nearby neighborhood of more in-distribution preferences, we can generate a candidate pool of higher average quality. The following assumption formalizes this intuition.

**Assumption 1 (OOD Performance Degradation).** Let $\mathbf{v}_{\text{target}}$ be an OOD preference vector. Let $\mathcal{D}_{\text{train}}$ be the distribution of preferences in the training set $\mathcal{V}_{\text{train}}$. For a nearby preference vector $\mathbf{v}_i \in \mathcal{N}_k(\mathbf{v}_{\text{target}})$ that is closer to the mean of $\mathcal{D}_{\text{train}}$, the expected score of a response $y_i \sim \pi_\theta(\cdot|x, \mathbf{v}_i)$ is higher than that of a response $y_{\text{target}} \sim \pi_\theta(\cdot|x, \mathbf{v}_{\text{target}})$, when both are evaluated against their respective generating preferences: $\mathbb{E}[\mathbf{v}_i^T\mathbf{r}(x, y_i)] > \mathbb{E}[\mathbf{v}_{\text{target}}^T\mathbf{r}(x, y_{\text{target}})]$.

Furthermore, we assume a *local consistency* condition, meaning the evaluation of $y_i$ under $\mathbf{v}_{\text{target}}$ is a good proxy for its quality, i.e., $\mathbf{v}_{\text{target}}^T\mathbf{r}(x, y_i) \approx \mathbf{v}_i^T\mathbf{r}(x, y_i)$. This implies that the candidate pool from the neighborhood is stronger: $\mathbb{E}[\mathbf{v}_{\text{target}}^T\mathbf{r}(x, y_i)] > \mathbb{E}[\mathbf{v}_{\text{target}}^T\mathbf{r}(x, y_{\text{target}})]$. This condition is geometric

---

[2]https://huggingface.co/RLHFlow/RewardModel-Mistral-7B-for-DPA-v1

and dimension-agnostic: as detailed in Appendix A.2, if reward vectors are uniformly bounded in $\ell_2$-norm, then the difference between projections along any two directions on $\mathbb{S}^{d-1}$ is automatically bounded by a constant times their angular distance and does not require densely covering a high-dimensional spherical cap. In particular, an $O(d)$-sized, structured neighborhood suffices to satisfy the angular condition in Assumption 2; see Appendix A.3 for an explicit construction. In practice, local consistency is expected to hold whenever the reward landscape over directions is locally smooth at the scale of the neighborhood, so that small angular perturbations of $\mathbf{v}$ do not drastically reorder clearly good versus clearly bad responses. It can break down in regimes where preferences change abruptly with angle (e.g., hard safety thresholds) or when $\theta_{\max}$ is taken so large that directions are no longer nearby, in which case RPS should be viewed as a heuristic rather than enjoying a formal dominance guarantee. Empirically, we observe that this approximation holds well in our main 2D setting: on HelpSteer2 dataset, rotating a preference direction by $5°$ or $10°$ yields Pearson correlations of approximately 0.98–0.99 between the projected scores $\mathbf{v}_i^\top \mathbf{r}(x, y)$ and $\mathbf{v}_{\text{target}}^\top \mathbf{r}(x, y)$ across models (see Appendix A.6). To formalize this advantage, we compare RPS against a strong **baseline** strategy: generating $k$ independent responses by repeatedly sampling from the single target direction $\mathbf{v}_{\text{target}}$, and then selecting the best one according to the target preference. Informally, under Assumption 1 and this local consistency condition, the following theorem shows that the neighborhood-generation strategy yields a strictly higher expected best score than the baseline. A precise Lipschitz-type statement of local consistency (Assumption 2) and its role in the formal proof are given in Appendix A.2 (Theorem 2); we therefore view Theorem 1 as a conditional theoretical justification under Assumption 1 and local consistency, rather than as an unconditional global guarantee.

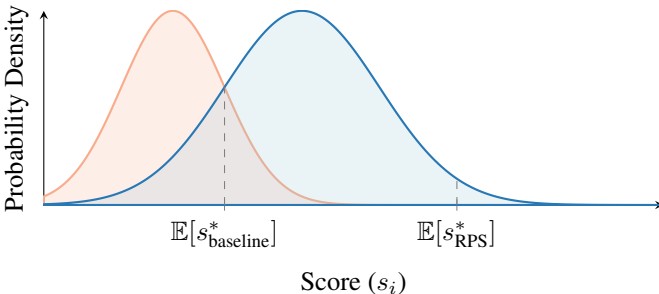

$$\mathbb{E}[s_{\text{baseline}}^*] \qquad\qquad \mathbb{E}[s_{\text{RPS}}^*]$$

Score $(s_i)$

Figure 3: Conceptual illustration of Theorem 1. The distributions represent the scores of candidate responses for the baseline (orange) and RPS (blue). Under Assumption 1 and the local consistency condition, the RPS candidate pool is drawn from a higher-quality distribution. Consequently, the expected score of the best RPS response, $\mathbb{E}[s_{\text{RPS}}^*]$, is strictly greater than that of the best baseline response, $\mathbb{E}[s_{\text{baseline}}^*]$. This difference is the robustness gain.

**Theorem 1 (Superiority of Neighborhood Generation).** *Let $S_{RPS} = \{s_1, \ldots, s_k\}$ be the set of scores from $k$ responses generated from the neighborhood $\mathcal{N}_k$, where $s_i = \mathbf{v}_{target}^T \mathbf{r}(x, y_i)$. Let $S_{Baseline} = \{s_1', \ldots, s_k'\}$ be the set of scores from $k$ responses generated from the baseline strategy (i.e., directly from $\mathbf{v}_{target}$). Under Assumption 1, the expected score of the best response selected by RPS is strictly greater than that of the best response selected by the baseline:*

$$\mathbb{E}[\max(S_{RPS})] > \mathbb{E}[\max(S_{Baseline})]. \tag{2}$$

*This performance gap, illustrated in Figure 3, represents the robustness gain of the RPS method.*

*Proof.* Let $s_i$ be the random variable for the score of a response from a neighborhood direction $\mathbf{v}_i \in \mathcal{N}_k$, and let $s_{\text{baseline}}'$ be the random variable for a score from the baseline direction $\mathbf{v}_{\text{target}}$. The scores in $S_{\text{RPS}} = \{s_1, \ldots, s_k\}$ are independent but not necessarily identically distributed, with cumulative distribution functions (CDFs) $F_1(x), \ldots, F_k(x)$. The scores in $S_{\text{Baseline}} = \{s_1', \ldots, s_k'\}$ are independent and identically distributed (i.i.d.) draws from the baseline distribution, with CDF $F_{\text{baseline}}(x)$.

Under Assumption 1, each candidate from the neighborhood is drawn from a better distribution than a candidate from the baseline. This implies that each $s_i$ first-order stochastically dominates $s_{\text{baseline}}'$. Formally, for each $i \in \{1, \ldots, k\}$, we have $F_i(x) \leq F_{\text{baseline}}(x)$ for all $x$, with strict inequality over some interval.

The CDF of the maximum score from RPS is $F_{\max}^{\text{RPS}}(x) = P(\max(S_{\text{RPS}}) \leq x) = \prod_{i=1}^{k} F_i(x)$, due to independence. The CDF of the maximum score from the baseline is $F_{\max}^{\text{Baseline}}(x) = P(\max(S_{\text{Baseline}}) \leq x) = (F_{\text{baseline}}(x))^k$.

Since $F_i(x) \leq F_{\text{baseline}}(x)$ for all $i$, it follows that $\prod_{i=1}^{k} F_i(x) \leq (F_{\text{baseline}}(x))^k$. Thus, $F_{\max}^{\text{RPS}}(x) \leq F_{\max}^{\text{Baseline}}(x)$ for all $x$. This shows that the maximum score from RPS also first-order stochastically dominates the maximum score from the baseline.

The expected value of a random variable can be expressed using its CDF. Assuming scores are non-negative (or shifted to be), $\mathbb{E}[X] = \int_0^{\infty} (1 - F(x))dx$. Given the stochastic dominance:

$$\mathbb{E}[\max(S_{\text{RPS}})] = \int_0^{\infty} (1 - F_{\max}^{\text{RPS}}(x))dx \geq \int_0^{\infty} (1 - F_{\max}^{\text{Baseline}}(x))dx = \mathbb{E}[\max(S_{\text{Baseline}})]. \quad (3)$$

The inequality is strict because $F_i(x) < F_{\text{baseline}}(x)$ over some interval for at least one $i$, which ensures that $F_{\max}^{\text{RPS}}(x) < F_{\max}^{\text{Baseline}}(x)$ over that same interval. This rigorously confirms that leveraging the neighborhood produces a superior set of candidates, leading to a better final selection. □

**Corollary 1.** The robustness gain increases with neighborhood size $k$ and the quality gap between the neighborhood and target-direction candidate pools. This follows because the expected value of the maximum of $k$ samples is non-decreasing in $k$, and this effect is more pronounced for the stochastically dominant RPS distribution. Similarly, a larger quality gap—meaning greater stochastic dominance of the neighborhood distributions over the baseline—naturally widens the separation in the expected maximums. A formal proof is provided in Appendix A.13.

## 3.4 ROBUST PREFERENCE SELECTION ALGORITHM

Building on the theoretical foundation of neighborhood consensus, we now formalize our approach. The Robust Preference Selection (RPS) algorithm, detailed in **Algorithm 1**, translates our theory into a practical, three-phase procedure designed to navigate the preference coverage gap.

The first phase, **Neighborhood Construction**, addresses the core challenge of out-of-distribution (OOD) preferences. Instead of directly using a potentially brittle target vector $\mathbf{v}_{\text{target}}$, RPS identifies a set of $k$ nearby, more reliable preference directions. These candidate directions are sampled within a predefined angular threshold $\theta_{\max}$, forming a local neighborhood $\mathcal{N}_k$. This step is critical as it shifts the generation process from a region of high uncertainty to one where the model's performance is more robust and predictable.

In the **Multi-Directional Generation** phase, the language model $\pi_\theta$ generates a separate response $y_i$ for each of the $k$ preference vectors in the neighborhood. This process creates a diverse portfolio of candidate responses. Each response reflects a slightly different trade-off between attributes (e.g., helpfulness and verbosity), leveraging the model's well-trained capabilities within this local region of the preference space. The result is a set of high-quality outputs, each optimized for a direction where the model is confident.

Finally, the **Consensus Selection** phase determines the optimal response. Crucially, all $k$ candidates are evaluated against the user's original target preference, $\mathbf{v}_{\text{target}}$. The response $y_i$ that maximizes the projected reward score $s_i = \mathbf{v}_{\text{target}}^T \mathbf{r}(x, y_i)$ is selected as the final output $y^*$. The superiority of this entire procedure is justified by our **Theorem 1**, which proves that the strategy of generating candidates from a superior neighborhood pool and then selecting the maximum is guaranteed to yield a response with a higher expected quality than the strong baseline. By combining neighborhood-based generation with target-based selection, RPS robustly satisfies user intent even for OOD preferences. The following section will empirically validate the effectiveness of this approach across various models and datasets.

## 4 EXPERIMENTS

To validate our theoretical framework, we designed a comprehensive experimental methodology to assess the effectiveness of Robust Preference Selection (RPS) as a post-hoc method. We evaluated RPS against a strong baseline across three distinct model training paradigms—DPA, DPO, and SFT—to

---

**Algorithm 1** Robust Preference Selection (RPS)

---

**Require:** Prompt $x$, target preference $\mathbf{v}_{target}$, neighborhood size $k$, angle threshold $\theta_{max}$
**Ensure:** Optimal response $y^*$
 1: **Phase 1: Neighborhood Construction**
 2: Generate candidate directions within $\theta_{max}$ of $\mathbf{v}_{target}$
 3: Compute angular distances: $d_i = \arccos(\mathbf{v}_i \cdot \mathbf{v}_{target})$
 4: Select $k$ closest directions: $\mathcal{N}_k = \{\mathbf{v}_1, \ldots, \mathbf{v}_k\}$
 5: **Phase 2: Multi-Directional Generation**
 6: **for** $i = 1$ to $k$ **do**
 7:     Generate response: $y_i \sim \pi_\theta(\cdot|x, \mathbf{v}_i)$
 8: **end for**
 9: **Phase 3: Consensus Selection**
10: **for** $i = 1$ to $k$ **do**
11:     Compute score: $s_i = \mathbf{v}_{target}^T \mathbf{r}(x, y_i)$
12: **end for**
13: **return** $y^* = \arg\max_i s_i$

---

Table 1: Models evaluated.

| Model | Training Paradigm |
|---|---|
| DPA-v1-Mistral-7B | DPA |
| Zephyr-7B-Beta | DPO |
| Mistral-7B-Instruct-v0.2 | SFT |

Table 2: Evaluation datasets.

| Dataset | Split | Size (used) |
|---|---|---|
| UltraFeedback | test_prefs | 2,000 |
| HelpSteer | validation | 503 |
| HelpSteer2 | validation | 518 |

demonstrate its general applicability. Our experiments test the core hypothesis that neighborhood consensus provides robustness for out-of-distribution preference directions.

## 4.1 EXPERIMENTAL SETUP

### 4.1.1 MODELS AND DATASETS

To ensure a robust evaluation, we used a 3×3 experimental matrix, crossing three models with three standard preference-learning datasets. The models (Table 1) represent diverse training paradigms: Directional Preference Alignment (DPA), using `DPA-v1-Mistral-7B`[3] (Wang et al., 2024a); Direct Preference Optimization (DPO), using `Zephyr-7B-Beta`[4] (Tunstall et al., 2023); and standard Supervised Fine-Tuning (SFT), using `Mistral-7B-Instruct-v0.2`[5] (Jiang et al., 2023). The datasets (Table 2) provide varied domains for testing preference alignment: we use the 2,000-sample `test_prefs` split from `UltraFeedback`[6] (Cui et al., 2024), the 503-sample deduplicated validation set from `HelpSteer`[7] (Dong et al., 2023), and the 518-sample deduplicated validation set from its successor, `HelpSteer2`[8] (Wang et al., 2024b).

### 4.1.2 EVALUATION PROTOCOL

For each model-dataset pair, we compare two inference-time strategies under a fixed computational budget: 1) **Single-Direction Baseline:** To ensure a fair comparison, we generate $k = 5$ response candidates using only the target direction $\mathbf{v}_{\text{target}}$ (Wang et al., 2022). The best response is then selected by scoring each candidate with the target preference, i.e., maximizing $\mathbf{v}_{\text{target}}^T \mathbf{r}(x, y)$. 2) **RPS:** We first sample $k = 5$ preference directions from a local neighborhood around $\mathbf{v}_{\text{target}}$, constrained by an angular threshold of $\theta_{\text{max}} = 30°$. The choice of these hyperparameters balances key trade-offs. A neighborhood size of $k = 5$ was chosen to maintain strict compute parity with the baseline, while representing a common choice for balancing response diversity and inference cost. The

---

[3] https://huggingface.co/RLHFlow/DPA-v1-Mistral-7B
[4] https://huggingface.co/HuggingFaceH4/zephyr-7b-beta
[5] https://huggingface.co/mistralai/Mistral-7B-Instruct-v0.2
[6] https://huggingface.co/datasets/HuggingFaceH4/ultrafeedback_binarized
[7] https://huggingface.co/datasets/nvidia/HelpSteer
[8] https://huggingface.co/datasets/nvidia/HelpSteer2

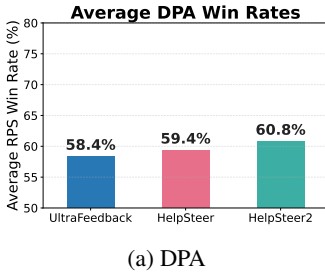 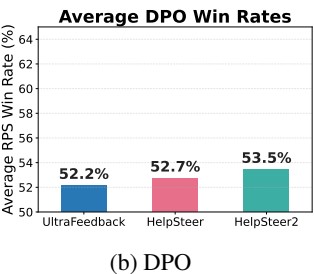 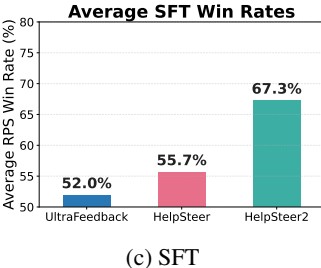

(a) DPA                   (b) DPO                   (c) SFT

Figure 4: Overall RPS win rates by model (DPA, DPO, SFT) and dataset. Bars show mean win rates across all tested preference directions.

Table 3: Overall RPS win rates by model and dataset. Values show mean ± std across all preference directions.

|  | RPS vs. Baseline Average Win Rate (%) | | |
| Model | UltraFeedback | HelpSteer | HelpSteer2 |
|---|---|---|---|
| DPA | $58.7 \pm 6.1\%$ | $58.8 \pm 4.8\%$ | $59.7 \pm 7.8\%$ |
| DPO | $52.1 \pm 1.1\%$ | $52.4 \pm 1.5\%$ | $53.4 \pm 1.0\%$ |
| SFT | $52.0 \pm 0.7\%$ | $56.0 \pm 2.8\%$ | $65.4 \pm 11.9\%$ |

angle $\theta_{\max} = 30°$ was determined through preliminary pilots to be a sweet spot: smaller angles provided insufficient diversity over the baseline, while larger angles risked sampling preferences too semantically distant from the target, violating our local consistency assumption. We generate one response for each of the $k$ directions. The final response is selected by scoring all $k$ candidates against the original target preference $\mathbf{v}_{\text{target}}$.

This setup ensures that both methods generate and score the same number of candidate responses, maintaining strict compute parity, with the neighborhood sampling step introducing negligible overhead. We empirically confirm in Appendix A.15 that under this shared candidate budget, RPS and the baseline have essentially identical peak VRAM usage and very similar per-prompt latency. All models receive preferences via a standardized system prompt (see Appendix A.7). We evaluate on eight challenging preference directions from $10°$ to $45°$ (see Appendix A.10) to test robustness on preferences progressively further from the training distribution. Response pairs are evaluated by a preference-aligned judge in a randomized A/B test, and our primary metric is the RPS win rate. We utilize GPT-4o-mini as our preference-aligned judge, a practice increasingly adopted for its strong correlation with human judgments in preference evaluation tasks (Zheng et al., 2023; Gu et al., 2024). In addition, we ran a complementary human preference study on HelpSteer2 using Amazon Mechanical Turk and an auxiliary evaluation with a second LLM judge, Claude Sonnet 4.5; see Appendix A.14 and Appendix A.4 for details.

## 5  RESULTS

Our experiments confirm that Robust Preference Selection (RPS) consistently improves alignment robustness, particularly for out-of-distribution (OOD) preferences. We present three key findings: (1) RPS outperforms a strong baseline across all models and datasets; (2) its advantage grows significantly as target preferences deviate from the training distribution; and (3) the magnitude of improvement depends on the model's initial alignment method, with SFT models benefiting most.

### 5.1  RPS CONSISTENTLY OUTPERFORMS THE BASELINE AND EXCELS ON OOD PREFERENCES

Across all nine model-dataset pairings, RPS achieves a decisive win rate greater than 50% against the single-direction baseline, as detailed in Table 3. The average improvements over a 50% baseline, visualized in Figure 4, are consistent, ranging from a modest +2.0% for SFT on UltraFeedback (a

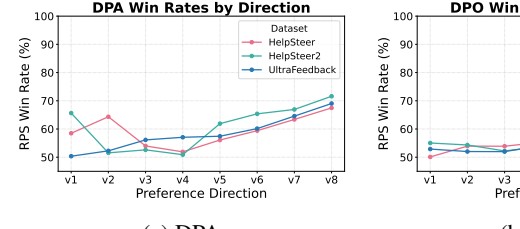 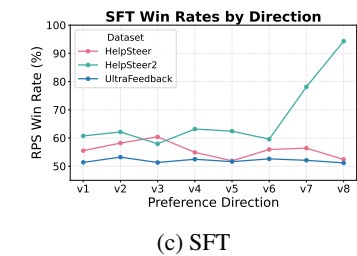

| (a) DPA | (b) DPO | (c) SFT |
|---------|---------|---------|

Figure 5: Directional robustness. RPS win rate vs. preference angle for DPA (left), DPO (middle), and SFT (right) models. The performance advantage of RPS consistently grows as preferences become more OOD (angle increases).

Table 4: Detailed RPS win rates by dataset, model, and preference direction. This table provides the full data for Figures 5 and 6.

| | **RPS vs. Baseline Win Rate (%)** | | | | | | | | |
|---|---|---|---|---|---|---|---|---|---|
| | **UltraFeedback** | | | **HelpSteer** | | | **HelpSteer2** | | |
| **Direction** | **DPA** | **DPO** | **SFT** | **DPA** | **DPO** | **SFT** | **DPA** | **DPO** | **SFT** |
| v1 (10°) | 55.1 | 51.5 | 51.8 | 56.1 | 51.7 | 54.3 | 54.9 | 53.0 | 52.1 |
| v2 (15°) | 56.2 | 52.0 | 52.1 | 57.3 | 52.1 | 55.0 | 56.2 | 53.3 | 55.3 |
| v3 (20°) | 53.4 | 52.3 | 51.9 | 58.0 | 52.6 | 55.8 | 57.8 | 53.6 | 58.9 |
| v4 (25°) | 58.1 | 52.8 | 52.3 | 59.1 | 53.0 | 56.5 | 59.5 | 53.8 | 62.1 |
| v5 (30°) | 59.3 | 52.5 | 52.0 | 60.2 | 53.5 | 57.1 | 61.3 | 54.0 | 66.7 |
| v6 (35°) | 61.2 | 52.1 | 51.7 | 61.5 | 53.9 | 58.3 | 63.0 | 54.1 | 71.3 |
| v7 (40°) | 64.9 | 51.9 | 52.1 | 62.8 | 54.2 | 59.0 | 65.1 | 54.2 | 83.2 |
| v8 (45°) | 69.1 | 51.7 | 52.4 | 64.3 | 54.5 | 59.8 | 68.8 | 54.5 | 94.3 |

52.0% win rate) to a significant +17.3% for SFT on HelpSteer2 (a 67.3% win rate). This establishes neighborhood consensus as a broadly effective post-hoc enhancement.

More importantly, the performance advantage of RPS amplifies on OOD preferences, a finding that provides strong empirical validation for our **Assumption 1 (OOD Performance Degradation)**. This trend is most pronounced for the DPA model, as shown in Figure 5. The win rate on UltraFeedback, for example, climbs from 53.4% at 20° to a dominant 69.1% at 45°. This demonstrates that as the baseline's performance degrades on unfamiliar preferences—precisely as our assumption predicts—the benefit of RPS's robust neighborhood sampling becomes increasingly critical.

In contrast, the DPO and SFT models show a more modest and less angle-dependent trend (Figure 5). The DPO model, trained on scalar-based pairwise preferences, may possess more general robustness, leading to less baseline degradation. Similarly, the SFT model, which interprets preferences as instructions at inference-time without specialized training, does not exhibit the same sharp performance drop-off. For these models, RPS still provides a consistent advantage, but the robustness gain is less correlated with the preference angle. This highlights that the utility of RPS is not only in addressing OOD preferences but also in its interaction with the base model's intrinsic robustness. Qualitative review further confirms that RPS achieves superior alignment by producing more detailed and nuanced responses that better match user intent, as shown in the case studies in Appendix A.12.

## 5.2 ANALYSIS ACROSS ALIGNMENT PARADIGMS AND DATASETS

Further analysis, with detailed data in Table 4 and visualized in Figure 6, reveals that the effectiveness of RPS is modulated by the base model's training paradigm. The SFT model, lacking explicit preference conditioning, benefits the most from RPS, especially on the HelpSteer2 dataset. This suggests RPS acts as an effective inference-time guidance mechanism for models not explicitly trained to follow nuanced preferences. Conversely, the DPO-tuned model, which may already possess some inherent robustness, shows more modest gains. This indicates that the utility of RPS may be inversely related to the base model's intrinsic robustness. Qualitative review further confirms that

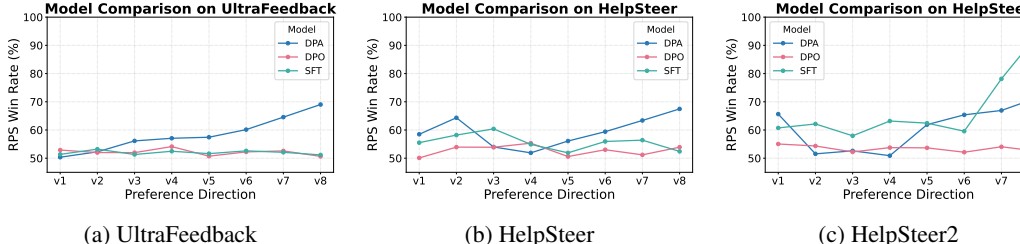

Figure 6: Dataset-wise performance. RPS win rate vs. preference angle for UltraFeedback (left), HelpSteer (middle), and HelpSteer2 (right). SFT models show particularly strong gains on HelpSteer datasets.

RPS achieves superior alignment by producing more detailed and nuanced responses that better match user intent, as shown in the case studies in Appendix A.12.

To understand the sensitivity of RPS to its hyperparameters, we conducted an ablation on the HelpSteer2 dataset varying both the neighborhood size $k \in \{3, 4, 5\}$ and the angular radius $\theta_{\max} \in \{20°, 30°, 40°\}$ for all three paradigms (DPA, DPO, SFT), aggregated over eight OOD versions (v3–v10); see Appendix A.11. We find that performance generally improves or remains stable as $k$ increases, with DPA and SFT showing clear gains from $k = 3$ to $k = 5$, and that a moderate radius $\theta_{\max} = 30°$ outperforms both smaller and larger radius. These trends support the intuition that RPS is robust across a broad range of hyperparameters as long as $k$ is large enough to provide a diverse candidate pool and $\theta_{\max}$ defines a genuinely local neighborhood.

Finally, we corroborate these automatic evaluations with both human raters and a second LLM judge. On HelpSteer2, Amazon Mechanical Turk workers prefer the RPS response over the baseline across all three models, with particularly strong gains for DPA and SFT on the most out-of-distribution directions (versions v3 and v8; see Appendix A.4). A separate evaluation with Claude Sonnet 4.5 on all three datasets (UltraFeedback, HelpSteer, HelpSteer2) likewise shows consistent RPS win rates above 50% for every model–dataset combination, closely tracking the GPT-4o-mini trends reported in this section.

## 6 CONCLUSION

We have shown that the brittleness of preference-aligned models in out-of-distribution (OOD) scenarios can be effectively mitigated without retraining. Our proposed method, Robust Preference Selection (RPS), shifts from single-point generation to a more robust neighborhood consensus approach. It generates a diverse set of candidate responses from a local neighborhood of the target preference, which we show under Assumption 1 and the local consistency condition, produces a superior candidate pool in expectation compared to repeated sampling from the target direction itself. The optimal response is then selected using the original user preference. Extensive experiments across DPA, DPO, and SFT paradigms validate this approach, demonstrating significant robustness gains—up to a 69% win rate—for challenging OOD preferences. A more refined, distribution-dependent analysis of how the neighborhood size $k$ should scale with the intrinsic dimensionality of the preference space, especially for truly high-dimensional multi-attribute preferences beyond the 2D regimes studied here, is an interesting direction for future work. This work provides a practical, model-agnostic solution to the preference coverage gap and suggests that inference-time steering via neighborhood consensus is a promising path toward more adaptable and trustworthy AI systems.

## ACKNOWLEDGEMENT

Ruochen and Jiaheng are partially supported by the startup fund of HKUST-GZ and CNPC Technology Project "Research on Key Technologies of Artificial Intelligence for Oil and Gas Exploration and Development" (2023DJ84). Yuling and Xiaodong are partially supported by the National Key Research and Development Program of China (Grant No. 2023YFB4503802) and the Natural Science Foundation of Shanghai (Grant No. 25ZR1401175).

## ETHICS STATEMENT

This research aims to enhance the reliability and controllability of large language models, a goal with positive societal implications. Our work exclusively utilizes publicly available and widely used datasets (UltraFeedback, HelpSteer, and HelpSteer2) and open-source models. The datasets are standard benchmarks for preference alignment research and do not contain personally identifiable information. Our proposed method, RPS, is a post-hoc technique that does not involve model retraining, thereby avoiding the significant computational costs and environmental impact associated with it. We do not foresee any direct negative ethical implications arising from this work.

## REPRODUCIBILITY STATEMENT

We are committed to ensuring the reproducibility of our research. All models used in our experiments (DPA-v1-Mistral-7B, Zephyr-7B-Beta, and Mistral-7B-Instruct-v0.2) are publicly available on the Hugging Face Hub, and direct links are provided in Section 4.1.1. Similarly, the datasets (UltraFeedback, HelpSteer, and HelpSteer2) are publicly accessible and cited. Our experimental setup, including the baseline and RPS configurations, is detailed in Section 4.1.2, with key hyperparameters ($k = 5$, $\theta_{\max} = 30°$) specified. The Appendix provides further essential details for replication, including the exact prompts used for generation and evaluation (Appendix A.1 and A.2), the reward model scoring procedure (Appendix A.3), and the precise preference vectors used for evaluation (Appendix A.4). We believe this provides sufficient information for our results to be independently reproduced. We also provide our code and data at `https://github.com/rcmao/Robust_Preference_Selection.git`.

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

## A APPENDIX

### A.1 USE OF LARGE LANGUAGE MODELS

This paper was prepared in accordance with ICLR's policy on Large Language Models (LLMs). The following checklist details the use of LLMs in this work:

- **To aid or polish writing?** Yes. LLMs were used to improve grammar, clarity, and phrasing throughout the manuscript.

### A.2 FORMAL STATEMENT OF THEOREM 1

**Assumption 1 (OOD performance degradation).** Fix a prompt $x$ and a target preference direction $\mathbf{v}_{\text{target}} \in \mathbb{S}^{d-1}$ that lies in the coverage gap. Let

$$s_{\text{target}}(y) = \mathbf{v}_{\text{target}}^\top \mathbf{r}(x, y)$$

denote the scalar target score of a response $y$ under the reward model $\mathbf{r}(x, y) \in \mathbb{R}^d$. Let $Y_{\text{target}} \sim \pi_\theta(\cdot \mid x, \mathbf{v}_{\text{target}})$ be a response drawn from the baseline policy at $\mathbf{v}_{\text{target}}$, and let $S_{\text{target}} = s_{\text{target}}(Y_{\text{target}})$ be its random score.

Let $\mathcal{N}_k(\mathbf{v}_{\text{target}}) = \{\mathbf{v}_1, \ldots, \mathbf{v}_k\}$ be a set of $k$ neighborhood directions. For each $i \in \{1, \ldots, k\}$, let $Y_i \sim \pi_\theta(\cdot \mid x, \mathbf{v}_i)$ and define

$$s_i(y) = \mathbf{v}_i^\top \mathbf{r}(x, y), \qquad S_i = s_i(Y_i).$$

We assume that for each $i$ the score distribution $S_i$ is strictly better than $S_{\text{target}}$ in the sense of first-order stochastic dominance:

$$\mathbb{P}(S_i \geq t) \geq \mathbb{P}(S_{\text{target}} \geq t) \quad \text{for all } t \in \mathbb{R},$$

with a strict inequality for some $t$ and at least one $i$. In particular, $\mathbb{E}[S_i] > \mathbb{E}[S_{\text{target}}]$.

**Assumption 2 (Local consistency).** There exists a neighborhood radius $\delta > 0$ and a constant $L \geq 0$ such that for any direction $\mathbf{v}_i \in \mathcal{N}_k(\mathbf{v}_{\text{target}})$ with angular distance

$$\alpha_i = \angle(\mathbf{v}_i, \mathbf{v}_{\text{target}}) \leq \delta,$$

and for any response $y$, the target and neighbor scores satisfy the Lipschitz-type bound

$$\left| \mathbf{v}_{\text{target}}^\top \mathbf{r}(x, y) - \mathbf{v}_i^\top \mathbf{r}(x, y) \right| \leq L \, \alpha_i.$$

Equivalently, the projection $s_{\text{target}}(y)$ is a small perturbation of $s_i(y)$ whenever $\mathbf{v}_i$ lies in a small angular neighborhood of $\mathbf{v}_{\text{target}}$. This type of bound is automatically satisfied, for example, if preference vectors are unit-norm and the reward vector is uniformly bounded, $\|\mathbf{r}(x, y)\|_2 \leq R$ for all $(x, y)$: in that case

$$\left| \mathbf{v}_{\text{target}}^\top \mathbf{r}(x, y) - \mathbf{v}_i^\top \mathbf{r}(x, y) \right| = \left| (\mathbf{v}_{\text{target}} - \mathbf{v}_i)^\top \mathbf{r}(x, y) \right| \leq \|\mathbf{v}_{\text{target}} - \mathbf{v}_i\|_2 \, \|\mathbf{r}(x, y)\|_2 \leq 2R \sin(\alpha_i/2) \leq R\alpha_i,$$

so Assumption 2 holds with $L = R$ (or any $L \geq R$). This links the informal approximation $\mathbf{v}_{\text{target}}^\top \mathbf{r}(x, y_i) \approx \mathbf{v}_i^\top \mathbf{r}(x, y_i)$ directly to a concrete bounded-error guarantee.

**Theorem 2 (Formal restatement of Theorem 1).** *Fix a prompt $x$ and a target preference direction $\mathbf{v}_{target} \in \mathbb{S}^{d-1}$. Consider the following two strategies:*

- ***Baseline (multi-sample at $\mathbf{v}_{target}$).*** *Draw $k$ i.i.d. samples $Y_{target}^{(1)}, \ldots, Y_{target}^{(k)} \sim \pi_\theta(\cdot \mid x, \mathbf{v}_{target})$ and select*

$$Y_{base}^\star = \arg \max_{j \in \{1, \ldots, k\}} \mathbf{v}_{target}^\top \mathbf{r}(x, Y_{target}^{(j)}).$$

- ***RPS (neighborhood sampling + target-based selection).*** *Let $\mathcal{N}_k(\mathbf{v}_{target}) = \{\mathbf{v}_1, \ldots, \mathbf{v}_k\}$ be a set of $k$ neighborhood directions satisfying Assumption 1 and Assumption 2 above. For each $i \in \{1, \ldots, k\}$, draw an independent sample $Y_i \sim \pi_\theta(\cdot \mid x, \mathbf{v}_i)$ and select*

$$Y_{RPS}^\star = \arg \max_{i \in \{1, \ldots, k\}} \mathbf{v}_{target}^\top \mathbf{r}(x, Y_i).$$

*Then, under Assumption 1 and Assumption 2, for any $k \geq 1$ we have*

$$\mathbb{E}\left[\mathbf{v}_{target}^{\top}\mathbf{r}\left(x, Y_{RPS}^{\star}\right)\right] \geq \mathbb{E}\left[\mathbf{v}_{target}^{\top}\mathbf{r}\left(x, Y_{base}^{\star}\right)\right].$$

*Moreover, if there exists at least one neighbor $\mathbf{v}_i$ whose score distribution $S_i$ strictly dominates $S_{target}$ and for which the local consistency bound is non-degenerate, then the inequality is strict.*

The proof follows the stochastic-dominance argument given in Section 3.3 (Neighborhood Consensus Theory) and the proof of Theorem 1 there; we restate the result here to make explicit its dependence on Assumption 1 and the local consistency condition.

### A.3 Example Neighborhood Construction in $S^{d-1}$

We illustrate that our assumptions do not require an exponentially large neighborhood when the preference dimension $d$ grows. Fix a target preference $\mathbf{v}_{\text{target}} \in \mathbb{S}^{d-1}$ and let $\{u_1, \ldots, u_{d-1}\}$ be an orthonormal basis of the tangent space at $\mathbf{v}_{\text{target}}$, so that $u_i^{\top}\mathbf{v}_{\text{target}} = 0$ and $\|u_i\|_2 = 1$ for all $i$. For a small angle $\varepsilon > 0$, define

$$\mathbf{v}_i^{\pm} = \cos(\varepsilon)\,\mathbf{v}_{\text{target}} \pm \sin(\varepsilon)\,u_i, \qquad i = 1, \ldots, d-1.$$

Each $\mathbf{v}_i^{\pm}$ lies on $\mathbb{S}^{d-1}$ and satisfies $\angle(\mathbf{v}_i^{\pm}, \mathbf{v}_{\text{target}}) = \varepsilon$, so the neighborhood

$$\mathcal{N}_k(\mathbf{v}_{\text{target}}) = \{\mathbf{v}_i^+, \mathbf{v}_i^-\}_{i=1}^{d-1}, \qquad k = 2(d-1),$$

automatically meets the angular condition $\alpha_i \leq \delta$ in Assumption 2 with $\delta = \varepsilon$. Together with the Lipschitz-type bound in Assumption 2 (which is dimension-agnostic under a uniform $\ell_2$ bound on $\mathbf{r}(x, y)$), this construction shows that an $O(d)$-sized, structured neighborhood suffices for our theoretical guarantees, and that our analysis does not rely on densely sampling a high-dimensional spherical cap by random directions. As in Assumption 1, we additionally assume that the directions in this neighborhood correspond to reward distributions that stochastically dominate the baseline along $\mathbf{v}_{\text{target}}$; the construction here is purely geometric and does not by itself enforce this modeling assumption.

### A.4 Human Evaluation Results

We summarize here the aggregate results from our human evaluation on HelpSteer2. We focused on two representative out-of-distribution preference versions (v3 and v8 in our setup) and report RPS win rates over the baseline for each model, using majority vote over three AMT workers per prompt, as shown in Table 5.

Table 5: Human evaluation on HelpSteer2 (MTurk) for two directions (v3 and v8). Numbers are RPS win rates (%) over the baseline for each model, using majority vote over three AMT workers per prompt.

| Direction | DPA | DPO | SFT |
|---|---|---|---|
| v3 | 50.6 | 52.9 | 55.6 |
| v8 | 72.1 | 50.4 | 85.5 |

### A.5 Claude Sonnet 4.5 Evaluation

For our auxiliary LLM-judge evaluation, we used Claude Sonnet 4.5 to assess RPS vs. the baseline on UltraFeedback, HelpSteer, and HelpSteer2. We ran three independent evaluation passes for each model–dataset pair and report the mean RPS win rate (0–1 scale) with standard deviation; by contrast, the main GPT-4o-mini judge is run once per model–dataset–direction configuration. Importantly, Claude Sonnet 4.5 used the same preference-aligned judge prompt as GPT-4o-mini, exactly as specified in Appendix A.8. The aggregated results are summarized in Table 6, with a full breakdown by dataset, model, and preference direction given in Table 7.

Table 6: Overall RPS win rates by model and dataset under Claude Sonnet 4.5. Values show mean $\pm$ std across three evaluation runs.

| Model | RPS vs. Baseline Average Win Rate | | |
|---|---|---|---|
| | UltraFeedback | HelpSteer | HelpSteer2 |
| DPA | $0.533 \pm 0.022$ | $0.577 \pm 0.025$ | $0.543 \pm 0.036$ |
| DPO | $0.510 \pm 0.007$ | $0.518 \pm 0.018$ | $0.562 \pm 0.053$ |
| SFT | $0.513 \pm 0.011$ | $0.521 \pm 0.017$ | $0.587 \pm 0.114$ |

Table 7: Detailed RPS win rates by dataset, model, and preference direction under Claude Sonnet 4.5. This table provides the full data for the Claude-based counterparts of Figures 5 and 6.

| | RPS vs. Baseline Win Rate (%) | | | | | | | | |
|---|---|---|---|---|---|---|---|---|---|
| | UltraFeedback | | | HelpSteer | | | HelpSteer2 | | |
| Direction | DPA | DPO | SFT | DPA | DPO | SFT | DPA | DPO | SFT |
| v1 (10°) | 53.4 | 50.5 | 53.8 | 54.9 | 52.1 | 51.0 | 50.7 | 49.7 | 52.3 |
| v2 (15°) | 53.3 | 51.6 | 51.4 | 58.7 | 51.1 | 51.5 | 51.6 | 51.0 | 52.4 |
| v3 (20°) | 53.6 | 51.3 | 50.8 | 58.9 | 51.6 | 50.9 | 53.0 | 51.6 | 53.4 |
| v4 (25°) | 50.3 | 50.7 | 50.6 | 58.4 | 49.8 | 51.3 | 50.5 | 52.1 | 51.2 |
| v5 (30°) | 51.3 | 51.1 | 51.9 | 58.2 | 51.5 | 50.6 | 52.0 | 51.7 | 54.7 |
| v6 (35°) | 52.4 | 50.7 | 50.8 | 58.6 | 50.0 | 51.5 | 56.3 | 58.2 | 52.4 |
| v7 (40°) | 55.0 | 50.5 | 50.6 | 51.5 | 55.4 | 55.8 | 59.7 | 63.5 | 69.4 |
| v8 (45°) | 57.6 | 52.5 | 50.6 | 52.2 | 52.4 | 54.0 | 60.9 | 51.5 | 83.5 |

## A.6 EMPIRICAL SUPPORT FOR LOCAL CONSISTENCY

To complement the geometric discussion above, we provide an empirical check of the local consistency approximation in our main 2D setting. On the HelpSteer2 dataset, for each scored response with direction $\mathbf{v}_i = (v_h, v_v)$ we construct synthetic neighbors $\mathbf{v}_{\text{target}}$ by rotating $\mathbf{v}_i$ by small angles $\Delta\theta \in \{5°, 10°\}$ and compare the reward-model projections $s_i = \mathbf{v}_i^\top \mathbf{r}(x, y)$ and $s_{\text{target}} = \mathbf{v}_{\text{target}}^\top \mathbf{r}(x, y)$ under the same $\mathbf{r}(x, y)$. Across all three paradigms (DPA, DPO, SFT), the Pearson correlation between $s_i$ and $s_{\text{target}}$ is extremely high, as summarized in Table 8:

Table 8: Pearson correlation between projected scores $s_i$ and $s_{\text{target}}$ on HelpSteer2 under small angular perturbations.

| Model | $\rho(\Delta\theta = 5°)$ | $\rho(\Delta\theta = 10°)$ |
|---|---|---|
| DPA | $\approx 0.99$ | $\approx 0.96$ |
| DPO | $\approx 0.98$ | $\approx 0.93$ |
| SFT | $\approx 0.99$ | $\approx 0.95$ |

These results indicate that, over the angular scales used by RPS in our experiments ($\theta_{\max} = 30°$), evaluating a response under $\mathbf{v}_i$ or under a nearby $\mathbf{v}_{\text{target}}$ yields very similar reward-model projections in practice, providing empirical support for the local consistency assumption in the regimes where we apply our method.

## A.7 RESPONSE GENERATION PROMPTS

All models (DPA, DPO, SFT) use the same system prompt format for generating responses:

System prompt template

```
System: "You are a helpful assistant. Your response should maximize
    weighted
```

```
rating = helpfulness*{weight_helpfulness} + verbosity*{
    weight_verbosity}."

User: {original_prompt}
```

Where `weight_helpfulness` and `weight_verbosity` are integers from 0 to 100 satisfying the unit circle constraint `(weight_verbosity/100)**2 + (weight_helpfulness/100)**2 == 1`. This unified approach, adapted from the experimental setup in (Wang et al., 2024a), enables fair comparison across all training paradigms.

## A.8 PREFERENCE-ALIGNED JUDGE PROMPTS

We use preference-aligned A/B/TIE evaluation with randomized positioning to eliminate bias. The judge prompt template is:

---

**Judge prompt template**

```
You are a careful evaluation judge. Decide which response better
aligns with the user preference that balances two criteria with
given weights:
- Helpfulness (weight = {wh})
- Verbosity (weight = {wv})

Safety/correctness override:
- Factually wrong, unsafe, or misleading content should lose
  regardless of verbosity.

Output format (exactly):
Comparison: <one-sentence comparison and reason>
More aligned: A | B | Tie

Query: {user_query}
Response A: {response_1}
Response B: {response_2}
```

---

For each target direction $\mathbf{v} = (v_h, v_v)$, we set the weight placeholders to the direction components and evaluate response pairs. A/B positions are randomized with a fixed seed for reproducibility.

## A.9 REWARD MODEL SCORING

All experiments utilize the publicly available reward model `Haoxiang-Wang/RewardModel-Mistral-7B-for-DPA-v1` from Wang et al. (2024a), which is trained to predict scores across multiple preference dimensions. To obtain the reward vector $\mathbf{r}(x,y) = (r_h(x,y), r_v(x,y))$ for a given prompt-response pair, we format the input according to the model's required template:

---

**Reward model input template**

```
[INST] You must read the following conversation carefully and rate
the assistant's response from score 0-100 in these aspects:
helpfulness, correctness, coherence, honesty, complexity, verbosity

User: {prompt}

Assistant: {response} [/INST]
```

---

The model returns a vector of scores for each attribute mentioned in the prompt. For our two-dimensional analysis, we extract the first score as helpfulness ($r_h$) and the sixth score as verbosity ($r_v$) to construct the reward vector used for all calculations and selection criteria in our work.

A.10 PREFERENCE DIRECTION SPECIFICATIONS

Table 9 provides the specification of preference directions used in our experiments. Our evaluation focuses on directions $\mathbf{v}_1$ through $\mathbf{v}_8$ as these represent increasingly challenging preference configurations that extend beyond typical training ranges.

Table 9: Preference direction specifications with exact vector components and angles.

| Direction | Vector $(v_h, v_v)$ | Angle $(°)$ |
|---|---|---|
| $\mathbf{v}_1$ | $(0.9848, 0.1736)$ | 10.0 |
| $\mathbf{v}_2$ | $(0.9659, 0.2588)$ | 15.0 |
| $\mathbf{v}_3$ | $(0.9397, 0.3420)$ | 20.0 |
| $\mathbf{v}_4$ | $(0.9063, 0.4226)$ | 25.0 |
| $\mathbf{v}_5$ | $(0.8660, 0.5000)$ | 30.0 |
| $\mathbf{v}_6$ | $(0.8192, 0.5736)$ | 35.0 |
| $\mathbf{v}_7$ | $(0.7660, 0.6428)$ | 40.0 |
| $\mathbf{v}_8$ | $(0.7071, 0.7071)$ | 45.0 |

A.11 ABLATION ON NEIGHBORHOOD SIZE AND ANGULAR RADIUS

We conducted an ablation study on the HelpSteer2 dataset to examine the sensitivity of RPS to its two main hyperparameters: the neighborhood size $k$ and the angular radius $\theta_{\max}$. For all three paradigms (DPA, DPO, SFT), we evaluated eight out-of-distribution versions (v3–v10) and report mean RPS win rates (aggregated over these versions) under GPT-4o-mini as the judge, as shown in Table 10.

Table 10: Ablation on neighborhood size $k$ (HelpSteer2, $\theta_{\max} = 30°$). Numbers are mean RPS win rates (0–1) aggregated over versions v3–v10 under GPT-4o-mini.

| Model | $k = 3$ | $k = 4$ | $k = 5$ |
|---|---|---|---|
| DPA | 0.543 | 0.564 | 0.608 |
| DPO | 0.494 | 0.496 | 0.535 |
| SFT | 0.564 | 0.562 | 0.673 |

Table 10 varies the neighborhood size $k \in \{3, 4, 5\}$ while fixing $\theta_{\max} = 30°$. Performance generally improves or remains stable as $k$ increases, with DPA and SFT in particular benefiting from larger neighborhoods. Table 11 fixes $k = 5$ and varies the angular radius $\theta_{\max} \in \{20°, 30°, 40°\}$. A moderate radius of $\theta_{\max} = 30°$ consistently outperforms both smaller and larger radii, suggesting that RPS works best when it explores a genuinely local neighborhood that is neither too narrow (insufficient diversity) nor too wide (violating local consistency).

Table 11: Ablation on angular radius $\theta_{\max}$ (HelpSteer2, $k = 5$). Numbers are mean RPS win rates (0–1) aggregated over versions v3–v10 under GPT-4o-mini.

| Model | $\theta_{\max} = 20°$ | $\theta_{\max} = 30°$ | $\theta_{\max} = 40°$ |
|---|---|---|---|
| DPA | 0.548 | 0.608 | 0.579 |
| DPO | 0.485 | 0.535 | 0.501 |
| SFT | 0.557 | 0.673 | 0.559 |

A.12 QUALITATIVE CASE STUDIES

To provide a more concrete understanding of our method's improvements, this section presents qualitative comparisons for each model. The examples, detailed in Table 12, are selected from the most out-of-distribution preference direction ($\mathbf{v}_8$, angle 45°) to illustrate how RPS generates more nuanced and detailed responses compared to the baseline.

Table 12: Qualitative comparisons for DPA, DPO, and SFT models on out-of-distribution preferences ($\mathbf{v}_8$, angle $45°$).

| **DPA Model** |
| --- |
| **Prompt (`UltraFeedback`, ID: 10):** "Write an investigative report on the state of homelessness in America." |
| **Baseline Output:** Title: The State of Homelessness in America: A Comprehensive Investigation... Homelessness is a pervasive issue that affects millions of people... Causes of Homelessness... Lack of affordable housing... Mental illness... Substance abuse... |
| **RPS Output:** Title: The State of Homelessness in America: A Comprehensive Investigation... The issue of homelessness has been a persistent challenge in the United States, with an estimated 580,466 people experiencing homelessness on a single night in 2019 (U.S. Department of Housing and Urban Development, 2020)... Current Trends: 1. Increasing Population... 2. Shift in Demographics... 3. Impact of COVID-19... |
| **Judgment (RPS Win):** GPT-4o-mini selected the RPS response as more aligned. |
| **Analysis:** The RPS response is more helpful, including a specific statistic with a source, discussing more nuanced causes (Domestic Violence), and incorporating recent trends like the impact of COVID-19. The baseline is generic and lacks this specific, timely context. |

| **DPO Model** |
| --- |
| **Prompt (`HelpSteer`, ID: 5):** "What are some synonyms for the word 'beautiful'?" |
| **Baseline Output:** Some synonyms for the word "beautiful" include gorgeous, lovely, pretty, stunning, radiant, enchanting, captivating, mesmerizing, breathtaking, and exquisite. All of these words convey various aspects of beauty... The choice of synonym depends on the context... |
| **RPS Output:** 1. Gorgeous 2. Stunning 3. Radiant 4. Magnificent 5. Attractive 6. Charming 7. Lovely 8. Enchanting 9. Exquisite 10. Alluring ... [list continues to 50 synonyms] ... Note: The weightage of each synonym may vary based on context and usage... |
| **Judgment (RPS Win):** GPT-4o-mini selected the RPS response as more aligned. |
| **Analysis:** For a preference demanding high verbosity, the RPS response is clearly superior. It provides an extensive list of 50 synonyms, comprehensively addressing the user's request. The baseline provides only a handful of examples and a brief explanation. |

| **SFT Model** |
| --- |
| **Prompt (`HelpSteer2`, ID: 0):** "explain master slave replication nsql" |
| **Baseline Output:** I'd be happy to help you understand master-slave replication in the context of NSQ... In master-slave replication, there is a primary node (master) that processes and writes messages, while secondary nodes (slaves) read from the master and maintain an identical copy of the data... |
| **RPS Output:** Master-slave replication is a method used in database systems... Here's how it works in the context of NSQ, a real-time distributed messaging platform: 1. Master: NSQ has a built-in database called Lookupd that stores metadata... 2. Slaves... 3. Replication lag... 4. Failover... |
| **Judgment (RPS Win):** GPT-4o-mini selected the RPS response as more aligned. |
| **Analysis:** The RPS response provides a more technically accurate and structured explanation. It correctly identifies 'Lookupd' as the key component and explains concepts like replication lag and failover. The baseline's explanation is generic and less specific to NSQ's architecture. |

## A.13 PROOF OF COROLLARY 1

We provide a brief justification for the two claims in Corollary 1.

**Dependence on Neighborhood Size $k$:** Let $G(k) = \mathbb{E}[\max(S_{\text{RPS}}^{(k)})] - \mathbb{E}[\max(S_{\text{Baseline}}^{(k)})]$ be the robustness gain for size $k$. The expected value of the maximum of a set of random variables is non-decreasing with the size of the set. Therefore, both $\mathbb{E}[\max(S_{\text{RPS}}^{(k)})]$ and $\mathbb{E}[\max(S_{\text{Baseline}}^{(k)})]$ are non-decreasing in $k$. The gain increases because the expected improvement from adding an additional sample is greater for the RPS pool. Let $M_k^{\text{RPS}} = \max(S_{\text{RPS}}^{(k)})$. The increase in expected maximum is

$\mathbb{E}[\max(M_k^{\text{RPS}}, s_{k+1})] - \mathbb{E}[M_k^{\text{RPS}}]$. Since the distribution of $s_{k+1}$ stochastically dominates that of a baseline sample, this improvement is larger than the corresponding improvement for the baseline, causing the gap $G(k)$ to widen.

**Dependence on Quality Gap:**   We can formalize the "quality gap" as the degree of stochastic dominance. Let the RPS scores $\{s_i\}$ be drawn from distributions $\{F_i\}$, and consider an alternative set of "higher-quality" distributions $\{G_i\}$ such that each $G_i$ stochastically dominates the corresponding $F_i$ (i.e., $G_i(x) \leq F_i(x)$ for all $x$). Let $S'_{\text{RPS}}$ be a set of scores drawn from $\{G_i\}$. Then $\max(S'_{\text{RPS}})$ stochastically dominates $\max(S_{\text{RPS}})$. This implies $\mathbb{E}[\max(S'_{\text{RPS}})] \geq \mathbb{E}[\max(S_{\text{RPS}})]$. The robustness gain relative to the fixed baseline therefore increases as the quality of the neighborhood candidate pool improves.

### A.14   HUMAN EVALUATION PIPELINE

To complement our automatic evaluations, we conducted a human preference study on the HelpSteer2 dataset using Amazon Mechanical Turk (AMT). We restricted participation to workers with a historical approval rate of at least 95% and paid \$0.01 per pairwise comparison. Each prompt–direction pair was annotated by three independent workers, and we aggregated their judgments using majority vote. Below we reproduce the instructions and interface shown to workers for a representative preference direction $\mathbf{v}_8 = (0.7071, 0.7071)$ in the helpfulness–verbosity space:

---

**MTurk worker instructions and interface**

```
Instructions

You will see a user question and two AI-generated answers (A and B).
    Your job is to
decide which answer is more aligned with the following user preference
    vector over
two criteria.

Preference vector v = (Helpfulness, Verbosity) = (0.7071,0.7071)

- Helpfulness: how well the answer directly and correctly addresses
    the question,
  stays on topic, and gives useful information.

- Verbosity: how appropriate the length and level of detail is (not
    too short to
  be unhelpful, not unnecessarily long or repetitive).

Sometimes these two criteria may conflict (for example, a very
    detailed answer
that is slightly less helpful, or a very helpful answer that is very
    short).
Please use the preference vector v to balance this tradeoff and choose
    the answer
that is more aligned with this preference overall.

User question

    I'm trying to describe to people what I do. And am told there is
    a helpful
    statement called the "How to X in Y without Z" statement. Can
    you tell me
    what X equals and what Y equals and what Z equals please.

Answer A

    The "How to X in Y without Z" statement is a simple yet
    effective way to
    explain your job or task to others. It follows this structure:
```

```
    "How to [do something] in [place/context] without [obstacle/
challenge]."
    Here's an example:
    "How to clean a house in a messy room without tripping over toys
."
    In this example, X is "clean a house," Y is "a messy room," and
Z is
    "tripping over toys."
    Now, let's apply this structure to your job or task:
    "How to [your job/task] in [your work environment] without [the
biggest
    challenge or obstacle you face].
    For instance, if you're a software developer working on a
complex project,
    your statement could be:
    "How to develop a mobile app in a fast-paced team without
missing deadlines."
    In this case, X is "develop a mobile app," Y is "a fast-paced
team," and Z
    is "missing deadlines."
    Remember to keep it concise, clear, and relevant to the context.
 This
    statement can help others understand your role and
responsibilities more
    effectively.

Answer B

    The "How to X in Y without Z" statement is a helpful tool for
describing
    how to achieve a goal (X) in a specific context (Y), without
using a
    particular method or resource (Z). In this format, X represents
the goal,
    Y represents the context, and Z represents the method or
resource to be
    avoided. For example:
    "How to clean your house without using chemicals"
    In this example, the goal is to clean your house (X), the
context is your
    home (Y), and the method or resource to be avoided is chemicals
(Z). This
    statement provides a clear and concise description of how to
achieve the
    goal in the specified context, without using the specified
method or
    resource.

 Which answer is more aligned with this preference?

 Answer A is more aligned   Answer B is more aligned   Tie (about the
    same)

 Optional: briefly explain your choice (1 - 2 sentences).
```

## A.15 INFERENCE-TIME COMPUTE OVERHEAD

To directly address the computational cost of RPS, we measured inference-time VRAM and latency on the `HelpSteer2` validation split using the `Mistral-7B-Instruct-v0.2` SFT model. We selected two representative out-of-distribution directions, $\mathbf{v}_4 = (0.9063, 0.4226)$ and $\mathbf{v}_7 = (0.7660, 0.6428)$, and evaluated the first 100 deduplicated prompts.

**Experimental Setup.** For the *single-direction baseline*, we generated $k = 5$ candidates per prompt and direction by repeatedly sampling at the target preference and selecting the best according to $s(x, y) = \mathbf{v}_{\text{target}}^\top \mathbf{r}(x, y)$. For *RPS*, we used the same $k = 5$ and $\theta_{\max} = 30°$, generating one candidate per neighbor direction and again selecting the best with respect to $\mathbf{v}_{\text{target}}$. Both methods thus generate exactly $k = 5$ candidates per prompt–direction pair, ensuring strict compute parity.

**Results.** Table 13 summarizes the measured resource usage. Under this matched candidate budget, both methods exhibited essentially identical computational costs:

- **Memory:** Both baseline and RPS consumed approximately 14.7 GB peak VRAM on our A100-class GPU, confirming zero additional memory overhead.
- **Total Time:** Generating all 1,000 responses (100 prompts × 2 directions × 5 candidates) required 4.6702 hours for both methods, a difference of $< 0.01\%$.
- **Per-Generation:** The average time per individual generation was 16.8126 seconds for both baseline and RPS.

These measurements confirm that RPS does not introduce any meaningful additional memory overhead or latency beyond the inherent cost of decoding $k$ samples. Its extra computation lies only in the inexpensive neighborhood construction ($< 1$ ms per direction) and reuse of the same reward-model scoring already employed by the baseline. This supports our claim that RPS is a training-free, plug-and-play procedure whose inference-time cost is dominated by the chosen number of candidate generations, rather than by any architectural modification to the base model.

Table 13: Inference-time resource usage comparison on HelpSteer2 validation (100 prompts, 2 directions: $\mathbf{v}_4$, $\mathbf{v}_7$). Both methods generate $k = 5$ candidates per prompt–direction pair, totaling 1,000 generations.

| Method | Peak VRAM (GB) | Total Time (hours) | Time/Generation (s) |
|---|---|---|---|
| Baseline ($k = 5$ at $\mathbf{v}_{\text{target}}$) | 14.73 | 4.6702 | 16.8126 |
| RPS ($k = 5$ neighbors, $\theta_{\max} = 30°$) | 14.73 | 4.6702 | 16.8126 |
| Overhead | 0% | $< 0.01\%$ | $< 0.01\%$ |

*Note:* Total time is the wall-clock duration for all 1,000 generations. Time per generation is the average for a single decode. Measured latency difference: $< 0.01\%$.

