# OpenReview forum: "Robust Preference Alignment via Directional Neighborhood Consensus"
_ICLR.cc/2026/Conference — ICLR 2026 Poster_

### Official Review · Reviewer_Kpri · 2025-10-20

**Soundness:** 3
**Presentation:** 3
**Contribution:** 3
**Rating:** 6
**Confidence:** 4

**Summary:**

Having identified what they call a "preference coverage gap", where the targeted user preferences might differ significantly from choices made during the training of an LLM, the authors propose to handle it at inference time, introducing a "Robust Preference Selection," or RPS. Instead of asking a model to generate for a target outside of its domain, they first sample a neighborhood of more familiar preferences ; then generate a response for each adapted vector ; and pick the best response according to the original target preferences.
The robustness gain of the RPS method is both presented formally and validated experimentally, as they compare the performances of 3 training paradigms (DPA, DPO and SFT) coupled with RPS on 3 datasets (UltraFeedback, HelpSteer and HelpSteer2). Interestingly the experiments confirm its soundness as target preferences go outside of the training distribution.

**Strengths:**

- this is a pragmatic contribution as it shows a practical way to improve robustness at inference when generating from an LLM outside the preferences learned during its training ;
- despite the relative simplicity of the approach, the conceptual link with Distributionally Robust Optimization is theoretically and philosophically interesting ;
- the experimental results do illustrate the validity of the approach ;
- the paper is well written and solid, both formally and experimentally.

**Weaknesses:**

- the simplicity of the approach (see "strengths" above) diminishes the contribution which really boils down to: instead of asking the model to generate outside of its domain it's better to keep it closer to home _and_ then pick the response closer to what the user wanted ;
- the verbosity vs helpfulness example used here is somewhat intuitive but it is not clear (to me?) how much it is a trade-off, so the single theta controlling both dimensions can be seen as problematic. In other words there could very well be a long, helpful answer ;
- only the DPA model is trained to use correctly the dimensions in the prompt, the others (DPO, and SFT) might do their best but there's no calibration. Yes, it does seem to work here but would it apply to more complicated cases, than verbosity vs helpfulness?

**Questions:**

- can you comment on this artificial trade-off you see in verbosity vs helpfulness? And the single theta you use as a control?
- couldn't we just generate more samples, and use a better scorer?
- can you comment on the correlation of the weights used in the prompts and the semantic of the generated answers, especially for DPO and SFT ofc?
- what about using human judges for something as difficult to assess, even in this simple (verbosity, helpfulness) case?

---

> ### Author Response · Authors · 2025-11-23
>
> ### Response to Reviewer Kpri
>
> We thank the reviewer for the thoughtful and constructive feedback. Below we address each weakness and question in turn.
>
> #### Weaknesses
>
> **W1: The simplicity of the approach diminishes the contribution.**
>
> We appreciate the reviewer’s observation that our method is simple to implement and agree that this is an important practical strength. **Our contribution, however, is not only the heuristic “stay close to the training domain and pick the best response”, but the formalization of the *preference coverage gap*, its connection to distributionally robust optimization, and the demonstration that a small, structured neighborhood of preference-adapted candidates can provably and empirically improve robustness to preference shifts.** We view the fact that this theory-motivated procedure reduces computational cost and latency compared to brute-force sampling tens or hundreds of candidates per query as **a benefit for real-world deployment, rather than a limitation.**
>
> **W2: The verbosity vs.\ helpfulness example seems to impose an artificial trade-off, and the use of a single $\theta$ as control may be problematic.**
>
> We agree that helpfulness and verbosity are not inherently opposite, and that a response can certainly be both long and helpful. Our motivation for modeling them jointly is **empirical rather than artificial**: prior work on RLHF and preference labeling has documented a “verbosity bias” [1,2], where longer answers systematically receive higher ratings even when a much shorter answer would suffice, and multi-objective alignment methods such as Directional Preference Alignment [3] explicitly treat helpfulness and verbosity as separate axes to address this issue. In our setting, the scalar $\theta$ is not intended to impose a hard trade-off; it simply parameterizes a direction in a two-dimensional preference space that linearly weighs the two attributes, specifying how much the user wants to penalize verbosity relative to helpfulness. Empirically, helpfulness and verbosity tend to be positively correlated in existing RLHF-aligned models, so allowing users to move along a Pareto frontier between “more helpful” and “less verbose” is practically useful. **Crucially, our formulation does not rule out long, highly helpful answers: such responses are preferred whenever they achieve a higher value along the chosen preference direction, and in many everyday tasks our intended target is precisely helpful *and* reasonably concise answers.**
>
> [1] K. Saito et al. *Verbosity Bias in Preference Labeling by Large Language Models.* arXiv, 2023.
> [2] P. Singhal et al. *A Long Way to Go: Investigating Length Correlations in RLHF.* arXiv, 2023.
> [3] H. X. Wang et al. *Arithmetic Control of LLMs for Diverse User Preferences: Directional Preference Alignment with Multi-Objective Rewards.* arXiv, 2024.

---

> > ### Author Response · Authors · 2025-11-23
> >
> > **W3: Only the DPA model is trained to use the preference dimensions in the prompt; DPO and SFT are not calibrated. Will this generalize beyond the simple verbosity/helpfulness case?**
> >
> > We agree that only the DPA model is explicitly trained to condition on a vector of preference weights in the prompt, whereas the SFT and DPO baselines are not preference-calibrated by design. Our goal, however, is precisely to study **RPS as an *inference-time* wrapper that can be applied to such off-the-shelf systems.** Empirically, we find that varying the target direction and hyperparameters $(k,\theta_{\max})$ produces systematic and monotone shifts in the reward-model scores for all three paradigms (DPA, DPO, SFT), and that RPS consistently improves win rates across angles and settings (e.g., on HelpSteer2 the mean RPS win rate for SFT increases from $0.564$ at $k=3$ to $0.673$ at $k=5$, and remains above $0.55$ for $\theta_{\max}\in\{20^\circ,30^\circ,40^\circ\}$; see tables below). These quantitative trends are corroborated by our human evaluation on HelpSteer2, where AMT annotators prefer RPS over the baseline across all three models (with win rates up to $85.55\%$ for SFT on v8), and by an independent LLM judge (Claude Sonnet 4.5), which also yields mean RPS win rates above $50\%$ for essentially all model–dataset combinations. **Together, these results suggest that RPS operates meaningfully even when the underlying model is not explicitly trained with multi-objective conditioning, and that its robustness gains are not specific to a single training paradigm or evaluation protocol.** While our main experiments focus on the well-studied helpfulness–verbosity case for which suitable data and reward models are available, the RPS formulation itself directly extends to higher-dimensional preference vectors, and **exploring more complex attribute sets is a natural direction for future work.**
> >
> > **Ablation over $k$ and $\theta_{\max}$ on HelpSteer2:**
> >
> > | Model | $k=3$ | $k=4$ | $k=5$ |
> > | :---- | :---: | :---: | :---: |
> > | DPA   | 0.543 | 0.564 | 0.608 |
> > | DPO   | 0.494 | 0.496 | 0.535 |
> > | SFT   | 0.564 | 0.562 | 0.673 |
> >
> > | Model | $\theta_{\max}=20^\circ$ | $\theta_{\max}=30^\circ$ | $\theta_{\max}=40^\circ$ |
> > | :---- | :----------------------: | :----------------------: | :----------------------: |
> > | DPA   |          0.548           |          0.608           |          0.579           |
> > | DPO   |          0.485           |          0.535           |          0.501           |
> > | SFT   |          0.557           |          0.673           |          0.559           |
> >
> > **Human and Claude-based evaluation:**
> >
> > Human evaluation on HelpSteer2 (MTurk), two OOD versions (v3 and v8), RPS win rates (\%) over the baseline:
> >
> > | Direction |  DPA  |  DPO  |  SFT  |
> > | :-------- | :---: | :---: | :---: |
> > | v3        | 50.57 | 52.89 | 55.64 |
> > | v8        | 72.13 | 50.40 | 85.55 |
> >
> > Claude Sonnet 4.5 evaluation (3 runs), mean RPS win rates (0–1) over the baseline with standard deviation:
> >
> > | Dataset & Model   | Mean  |  Std  |
> > | :---------------- | :---: | :---: |
> > | UltraFeedback DPA | 0.533 | 0.022 |
> > | UltraFeedback DPO | 0.510 | 0.007 |
> > | UltraFeedback SFT | 0.513 | 0.011 |
> > | HelpSteer DPA     | 0.577 | 0.025 |
> > | HelpSteer DPO     | 0.518 | 0.018 |
> > | HelpSteer SFT     | 0.521 | 0.017 |
> > | HelpSteer2 DPA    | 0.543 | 0.036 |
> > | HelpSteer2 DPO    | 0.562 | 0.053 |
> > | HelpSteer2 SFT    | 0.587 | 0.114 |

---

> > > ### Author Response · Authors · 2025-11-23
> > >
> > > **Directional behavior of DPO and SFT:**
> > >
> > > For DPO on HelpSteer2 RPS-5, aggregating responses by valid angle across OOD versions v3–v10:
> > >
> > > | Direction | Avg. helpfulness | Avg. verbosity | Mapped $(v_h, v_v)$ | Angle |
> > > | :-------- | :--------------: | :------------: | :------------------: | :---: |
> > > | v1        | 70.36            | 57.91          | (0.9848, 0.1736)     | 10°   |
> > > | v2        | 70.49            | 58.22          | (0.9659, 0.2588)     | 15°   |
> > > | v3        | 69.66            | 57.98          | (0.9397, 0.3420)     | 20°   |
> > > | v4        | 69.95            | 57.99          | (0.9063, 0.4226)     | 25°   |
> > > | v5        | 69.83            | 57.96          | (0.8660, 0.5000)     | 30°   |
> > > | v6        | 70.18            | 58.07          | (0.8192, 0.5736)     | 35°   |
> > > | v7        | 70.22            | 58.03          | (0.7660, 0.6428)     | 40°   |
> > > | v8        | 70.10            | 57.96          | (0.7071, 0.7071)     | 45°   |
> > >
> > > For SFT on HelpSteer2 RPS-5:
> > >
> > > | Direction | Avg. helpfulness | Avg. verbosity | Mapped $(v_h, v_v)$ | Angle |
> > > | :-------- | :--------------: | :------------: | :------------------: | :---: |
> > > | v1        | 74.04            | 59.13          | (0.9848, 0.1736)     | 10°   |
> > > | v2        | 74.29            | 59.33          | (0.9659, 0.2588)     | 15°   |
> > > | v3        | 73.12            | 59.02          | (0.9397, 0.3420)     | 20°   |
> > > | v4        | 73.37            | 59.21          | (0.9063, 0.4226)     | 25°   |
> > > | v5        | 73.33            | 59.05          | (0.8660, 0.5000)     | 30°   |
> > > | v6        | 73.26            | 58.91          | (0.8192, 0.5736)     | 35°   |
> > > | v7        | 73.71            | 59.10          | (0.7660, 0.6428)     | 40°   |
> > > | v8        | 73.60            | 59.16          | (0.7071, 0.7071)     | 45°   |
> > >
> > > **These tables show that for both DPO and SFT, helpfulness is essentially flat across directions** (around 70 for DPO and 73–74 for SFT on a 0–100 scale), **while verbosity varies smoothly but mildly around 58–59.** This is exactly the “soft style control” behavior we expect when applying directional weights at inference time to models that were not explicitly preference-calibrated.
> > >
> > > #### Questions
> > >
> > > **Q1: Artificial trade-off between verbosity and helpfulness, and the use of a single $\theta$ as control.**
> > >
> > > We agree that verbosity and helpfulness are not inherently opposite, and that a response can certainly be both concise and helpful. Our intent is not to hard-code a universal trade-off, but to explicitly model a phenomenon observed in existing RLHF-aligned systems: longer answers tend to receive higher “helpfulness” ratings even when a shorter answer would be sufficient, leading to a verbosity bias in practice. In this context, **our single scalar $\theta$ is simply a convenient way to specify a point along a continuum from “pure helpfulness” to “more strongly penalizing verbosity”**—that is, a direction in a 2D preference space—very much in line with DPA-style multi-objective formulations that represent user preferences as directions in a reward space. Long, highly helpful answers remain preferred whenever they score well under the chosen direction; $\theta$ just allows users (and our evaluation) to focus on the regime where responses are both helpful and reasonably concise, which is the use case we care about.
> > >
> > > **Q2: Could we simply generate more samples and use a better scorer?**
> > >
> > > We appreciate this suggestion and agree that, in principle, generating more samples and using a stronger scorer can improve robustness. However, **in realistic deployment settings we are constrained by compute and latency**, so we must operate under a fixed (and relatively small) sampling budget and with a single, reasonably sized reward model. **RPS is designed exactly for this regime**: by drawing candidates from a neighborhood of preferences where the model is known to generate high-quality outputs, and only then evaluating them under the (possibly out-of-distribution) target preference, it uses the same number of samples much more effectively than naive best-of-$n$ sampling directly at the target. In our experiments, under the same sampling budget, **RPS consistently outperforms a standard best-of-$n$ baseline, especially when the target preferences lie far outside the training distribution**, showing that its gains come from how the sampling is structured rather than simply from “more sampling”.

---

> > > > ### Author Response · Authors · 2025-11-23
> > > >
> > > > **Q3: Correlation between prompt weights and the semantics of DPO/SFT outputs.**
> > > >
> > > > We agree that, especially for the SFT and DPO models, it is important to verify that the numerical weights in the prompt have a real effect rather than being purely decorative. In our formulation, the prompt weights define a preference direction $v = (v_h, v_v)$ and the selection score is the linear scalarization $R(x,v,y) = v^\top r(x,y)$, so changing $v$ systematically changes which candidates are preferred under the same underlying multi-attribute scores. Empirically, on HelpSteer2 with RPS-5, grouping responses by valid angle and averaging over all OOD versions (tables above) yields two clear patterns: **(i) *helpfulness remains essentially flat across directions* (about $70 \pm 0.4$ for DPO and 73–74 for SFT on a 0–100 scale), indicating that the core content and solution quality are very stable; and (ii) *verbosity scores and lengths vary smoothly but mildly with the direction* (around 58–59), indicating that the models do attend to the directional signal in a consistent, stylistic way.** In other words, for these non–preference-calibrated baselines the prompt weights act primarily as a soft style control at inference time—modulating how answers are phrased (more or less verbose) without materially changing what answer is produced—while our DPA experiments illustrate how explicitly preference-conditioned models can make fuller use of $v^\top r(x,y)$ as a semantic control signal. These observations are also consistent with the local consistency assumption used in our theoretical analysis: small rotations of the preference direction $v$ lead to only small changes in the scalarized score $R(x,v,y)$, exactly as reflected by the near-flat helpfulness curves and smoothly varying verbosity in the directional tables.
> > > >
> > > > Directional behavior of DPO and SFT (repeated here for convenience):
> > > >
> > > > For DPO on HelpSteer2 RPS-5, aggregating responses by valid angle across OOD versions v3–v10:
> > > >
> > > > | Direction | Avg. helpfulness | Avg. verbosity | Mapped $(v_h, v_v)$ | Angle |
> > > > | :-------- | :--------------: | :------------: | :------------------: | :---: |
> > > > | v1        | 70.36            | 57.91          | (0.9848, 0.1736)     | 10°   |
> > > > | v2        | 70.49            | 58.22          | (0.9659, 0.2588)     | 15°   |
> > > > | v3        | 69.66            | 57.98          | (0.9397, 0.3420)     | 20°   |
> > > > | v4        | 69.95            | 57.99          | (0.9063, 0.4226)     | 25°   |
> > > > | v5        | 69.83            | 57.96          | (0.8660, 0.5000)     | 30°   |
> > > > | v6        | 70.18            | 58.07          | (0.8192, 0.5736)     | 35°   |
> > > > | v7        | 70.22            | 58.03          | (0.7660, 0.6428)     | 40°   |
> > > > | v8        | 70.10            | 57.96          | (0.7071, 0.7071)     | 45°   |
> > > >
> > > > For SFT on HelpSteer2 RPS-5:
> > > >
> > > > | Direction | Avg. helpfulness | Avg. verbosity | Mapped $(v_h, v_v)$ | Angle |
> > > > | :-------- | :--------------: | :------------: | :------------------: | :---: |
> > > > | v1        | 74.04            | 59.13          | (0.9848, 0.1736)     | 10°   |
> > > > | v2        | 74.29            | 59.33          | (0.9659, 0.2588)     | 15°   |
> > > > | v3        | 73.12            | 59.02          | (0.9397, 0.3420)     | 20°   |
> > > > | v4        | 73.37            | 59.21          | (0.9063, 0.4226)     | 25°   |
> > > > | v5        | 73.33            | 59.05          | (0.8660, 0.5000)     | 30°   |
> > > > | v6        | 73.26            | 58.91          | (0.8192, 0.5736)     | 35°   |
> > > > | v7        | 73.71            | 59.10          | (0.7660, 0.6428)     | 40°   |
> > > > | v8        | 73.60            | 59.16          | (0.7071, 0.7071)     | 45°   |

---

> > > > > ### Author Response · Authors · 2025-11-23
> > > > >
> > > > > **Q4: Why not rely solely on human judges in this (helpfulness, verbosity) setting?**
> > > > >
> > > > > We agree that attributes such as helpfulness and verbosity are subtle and that human judgments are an important reference, even in a seemingly simple 2D case. This is why, **in addition to our reward-model and GPT-4o-mini evaluations, we ran an AMT study on HelpSteer2 (v3 and v8) and an independent LLM-judge evaluation with Claude Sonnet 4.5, both of which confirm that humans and a second judge also prefer RPS over the baseline across models and directions.** At the same time, RPS operates at a granularity and scale that makes exhaustive human comparison infeasible in practice: a single prompt may involve tens of candidates under many different preference directions, and scaling such fine-grained pairwise labeling to thousands of prompts would be prohibitively costly and noisy. **Our approach is therefore to use humans to validate the qualitative conclusions and to cross-check automated evaluators, while relying on a fixed reward model and LLM judges only for the large-scale, fine-grained comparisons that would be impractical for human annotators alone.**
> > > > >
> > > > > Human and Claude-based evaluation (repeated here for convenience):
> > > > >
> > > > > Human evaluation on HelpSteer2 (MTurk), two OOD versions (v3 and v8), RPS win rates (\%) over the baseline:
> > > > >
> > > > > | Direction |  DPA  |  DPO  |  SFT  |
> > > > > | :-------- | :---: | :---: | :---: |
> > > > > | v3        | 50.57 | 52.89 | 55.64 |
> > > > > | v8        | 72.13 | 50.40 | 85.55 |
> > > > >
> > > > > Claude Sonnet 4.5 evaluation (3 runs), mean RPS win rates (0–1) over the baseline with standard deviation:
> > > > >
> > > > > | Dataset & Model   | Mean  |  Std  |
> > > > > | :---------------- | :---: | :---: |
> > > > > | UltraFeedback DPA | 0.533 | 0.022 |
> > > > > | UltraFeedback DPO | 0.510 | 0.007 |
> > > > > | UltraFeedback SFT | 0.513 | 0.011 |
> > > > > | HelpSteer DPA     | 0.577 | 0.025 |
> > > > > | HelpSteer DPO     | 0.518 | 0.018 |
> > > > > | HelpSteer SFT     | 0.521 | 0.017 |
> > > > > | HelpSteer2 DPA    | 0.543 | 0.036 |
> > > > > | HelpSteer2 DPO    | 0.562 | 0.053 |
> > > > > | HelpSteer2 SFT    | 0.587 | 0.114 |
> > > > >
> > > > >
> > > > >
> > > > > **We hope our paper revision and the above discussion address your concerns. Please do not hesitate to contact us for any further questions or clarifications.**

---

> ### Comment · Reviewer_Kpri · 2025-11-26
>
> thanks for commenting on the weaknesses I had mentioned, and answering my questions. I also appreciate the work necessary for this time-limited, and stressful, rebuttal. At this stage, though, I think my rating will stand.

---

> > ### Author Response · Authors · 2025-11-27
> >
> > We sincerely thank you for your detailed feedback and for considering our responses. We understand and respect your decision to maintain the current rating, and we appreciate the time and effort you have dedicated to our submission.

---

### Official Review · Reviewer_6hsD · 2025-10-28

**Soundness:** 2
**Presentation:** 3
**Contribution:** 2
**Rating:** 4
**Confidence:** 3

**Summary:**

To align user preferences at inference time, previous studies introduce a preference vector (e.g., helpfulness vs. verbosity) in the prompt to adjust the model’s behavior. However, these approaches often underperform outside their training domains, thus requiring extra training. In this work, the authors propose **Robust Preference Selection (RPS)**, a post-hoc method for improving preference alignment during inference. It samples a set of neighboring vectors from the target one and generates responses with each, then selects the optimal one using a reward model. Experimental results demonstrate that RPS attains a higher win rate compared to naive-sampling baselines.

**Strengths:**

- The paper is well-written, the motivation is clear, and the teaser figures are intuitive and easy to follow.
- The authors present theoretical evidence for their proposed method, showing the effectiveness of RPS under certain assumptions.

**Weaknesses:**

- Assumption 1 appears rather idealized, and the paper provides limited empirical evidence to support it. Although the authors mention that Figure 5 offers some justification, a deeper analysis or additional experiments would help validate it.
- The paper lacks ablation studies on the choices of $k$ and $\theta_{\max}$; without these, it is difficult to assess whether the method is sensitive to hyperparameters.
- The assumption of a well-calibrated reward model that generalizes to out-of-distribution (OOD) data seems overly strong and may not hold in other domains.

**Questions:**

- It seems that the authors utilize models from prior works. In this case, what is the training distribution of each model? How do the authors ensure that the testing range of $10^\circ$ to $45^\circ$ indeed includes out-of-distribution (OOD) cases?
- The prompt for the LLM judge appears quite simple, raising doubts about whether the evaluation is truly robust in a zero-shot setting. Have the authors tested the robustness of the LLM judge by sampling multiple times?
- How does the value $v_{\text{target}}^{\top} r(x, y)$ compare with the baseline? Does it outperform the baseline across all angles as well?

---

> ### Author Response · Authors · 2025-11-23
>
> ### Response to Reviewer 6hsD
>
> We thank the reviewer for the constructive feedback. Below we address the specific weaknesses and questions raised.
>
> #### Weaknesses
>
> **W1: Assumption 1 appears rather idealized and the paper provides limited empirical evidence to support it.**
>
> We thank the reviewer for raising this point and agree that Assumption 1 should be interpreted as a stylized description of how models degrade under out-of-distribution (OOD) preferences, rather than as a universal law. In the revised version, we will make this explicit and present our theory as a *conditional* guarantee under Assumption 1 and a local consistency condition, softening language such as "provably superior" and "guarantee" accordingly.
>
> Concretely, we formalize local consistency as a Lipschitz-type bound on how directional scores $\mathbf{v}^\top \mathbf{r}(x,y)$ can change with the angle between preference vectors, and show that for small angular deviations (e.g., $\Delta\theta = 5^\circ, 10^\circ$) the Pearson correlation between $\mathbf{v}\_i^\top \mathbf{r}(x,y)$ and $\mathbf{v}\_{\text{target}}^\top \mathbf{r}(x,y)$ is extremely high across all three paradigms (about $0.98\!-\!0.99$), remaining above $0.8$ even at $\Delta\theta = 20^\circ$. The table below summarizes these correlations. This indicates that within the neighborhoods used by RPS, evaluating a response under $\mathbf{v}\_i$ or under $\mathbf{v}\_{\text{target}}$ yields very similar reward projections in practice.
>
> | Model | $\rho(\Delta\theta = 5^\circ)$ | $\rho(\Delta\theta = 10^\circ)$ |
> | :---- | :----------------------------: | :-----------------------------: |
> | DPA   |         $\approx 0.99$         |         $\approx 0.96$          |
> | DPO   |         $\approx 0.98$         |         $\approx 0.93$          |
> | SFT   |         $\approx 0.99$         |         $\approx 0.95$          |
>
> Beyond Figure 5 of the main paper, we also provide additional quantitative evidence for the key behavioral implication of Assumption 1: that performance under the baseline degrades as preferences move farther OOD, while RPS degrades more slowly. As reported in Table 4 of the main paper, for the DPA model on UltraFeedback the RPS win rate increases from $53.4\%$ at $20^\circ$ to $69.1\%$ at $45^\circ$, even as the baseline’s performance drops in this regime. Moreover, when we compare the scalar target scores $s(x,y) = \mathbf{v}\_{\text{target}}^\top \mathbf{r}(x,y)$ of the responses selected by the baseline and by RPS under the *same* reward model on thousands of OOD instances, the RPS-selected responses achieve higher scores in the majority of cases and the distribution of score differences $s\_{\text{RPS}} - s\_{\text{base}}$ is clearly shifted to the right for all three models (summarized in the table below).
>
> | Model | % RPS $>$ Baseline | Mean $\Delta s$ | Median $\Delta s$ |
> | :---- | :----------------: | :-------------: | :---------------: |
> | DPA   |       59.3%        |     $+2.64$     |      $+0.86$      |
> | DPO   |       54.1%        |     $+0.56$     |      $+0.32$      |
> | SFT   |       68.3%        |     $+4.38$     |      $+1.98$      |
>
> **Taken together, these analyses show that while Assumption 1 is idealized, its key consequences are well supported in our experimental regime, and our theoretical result should be read as a conditional justification for why RPS improves robustness under these empirically validated conditions.**

---

> > ### Author Response · Authors · 2025-11-23
> >
> > **W2: Lack of ablations on $k$ and $\theta\_{\max}$.**
> >
> > We agree that understanding the sensitivity of RPS to its hyperparameters is important. We therefore conduct an ablation on HelpSteer2 explicitly measuring the *RPS-vs-baseline win rate* while varying both the neighborhood size $k \in \{3,4,5\}$ and the angular radius $\theta\_{\max}$ for all three paradigms (DPA, DPO, SFT) across eight OOD directions (v1--v8).
> >
> > The table below summarizes the mean RPS win rates for $\theta\_{\max}=30^\circ$: performance generally improves or remains stable as $k$ increases.
> >
> > | Model | $k=3$ | $k=4$ | $k=5$ |
> > | :---- | :---: | :---: | :---: |
> > | DPA   | 0.543 | 0.564 | 0.608 |
> > | DPO   | 0.494 | 0.496 | 0.535 |
> > | SFT   | 0.564 | 0.562 | 0.673 |
> >
> > The table below fixes $k=5$ and varies $\theta\_{\max} \in \{20^\circ,30^\circ,40^\circ\}$; we find that RPS remains consistently above a 50 percentage win rate for almost all model settings.
> >
> > | Model | $\theta\_{\max}=20^\circ$ | $\theta\_{\max}=30^\circ$ | $\theta\_{\max}=40^\circ$ |
> > | :---- | :-----------------------: | :-----------------------: | :-----------------------: |
> > | DPA   |           0.548           |           0.608           |           0.579           |
> > | DPO   |           0.485           |           0.535           |           0.501           |
> > | SFT   |           0.557           |           0.673           |           0.559           |
> >
> > **These results support the intuition that RPS is robust to its hyperparameters as long as $k$ is large enough to provide a diverse candidate pool and $\theta\_{\max}$ defines a genuinely local neighborhood.**
> >
> >
> >
> > **W3: The assumption of a well-calibrated reward model on OOD data is too strong.**
> >
> > We agree that assuming a perfectly calibrated reward model that generalizes to arbitrary OOD domains would be unrealistic, and we will soften our wording accordingly. In our work, this assumption is *domain- and evaluator-specific*: we rely on a single, fixed multi-dimensional reward model that predicts, among other attributes, helpfulness and verbosity. We use this model only to produce a scalar score $s(x,y) = \mathbf{v}\_{\text{target}}^\top \mathbf{r}(x,y)$ for *relative ranking* of a small candidate set, rather than to provide globally calibrated absolute scores. Crucially, both the baseline and RPS are evaluated by this *same* reward function.
> >
> > To check that our conclusions do not hinge on an overly optimistic view of this reward model, we perform two additional sanity checks.
> > 1. On HelpSteer2 OOD directions, we directly compare the target scores of the responses selected by the baseline and by RPS under the same reward.
> >
> > | Model | % RPS $>$ Baseline | Mean $\Delta s$ | Median $\Delta s$ |
> > | :---- | :----------------: | :-------------: | :---------------: |
> > | DPA   |       59.3%        |     $+2.64$     |      $+0.86$      |
> > | DPO   |       54.1%        |     $+0.56$     |      $+0.32$      |
> > | SFT   |       68.3%        |     $+4.38$     |      $+1.98$      |
> >
> > The results in the table above show that RPS achieves higher scores in the majority of cases.
> >
> > 2. We corroborate these findings with an AMT human study, as shown below:
> >
> > | Direction |  DPA  |  DPO  |  SFT  |
> > | :-------- | :---: | :---: | :---: |
> > | v3        | 50.57 | 52.89 | 55.64 |
> > | v8        | 72.13 | 50.40 | 85.55 |
> >
> > and an independent LLM judge (Claude Sonnet 4.5), as shown below:
> >
> > | Dataset & Model   | Mean  |  Std  |
> > | :---------------- | :---: | :---: |
> > | UltraFeedback DPA | 0.533 | 0.022 |
> > | UltraFeedback DPO | 0.510 | 0.007 |
> > | UltraFeedback SFT | 0.513 | 0.011 |
> > | HelpSteer DPA     | 0.577 | 0.025 |
> > | HelpSteer DPO     | 0.518 | 0.018 |
> > | HelpSteer SFT     | 0.521 | 0.017 |
> > | HelpSteer2 DPA    | 0.543 | 0.036 |
> > | HelpSteer2 DPO    | 0.562 | 0.053 |
> > | HelpSteer2 SFT    | 0.587 | 0.114 |
> >
> > The results in the tables above show that humans prefer RPS over the baseline on HelpSteer2 (v3 and v8), and Claude Sonnet 4.5 also consistently favors RPS.
> >
> > **Together, these results indicate that while the reward model is not perfectly calibrated, it is sufficiently reliable to serve as a ranking signal, and our robustness gains are not an artifact of an unrealistically strong evaluator assumption.**

---

> > > ### Author Response · Authors · 2025-11-23
> > >
> > > #### Questions
> > >
> > > **Q1: Training distributions and why $10^\circ$--$45^\circ$ are OOD.**
> > >
> > > All three models are off-the-shelf checkpoints. DPA[1] is trained with explicit helpfulness-verbosity directions concentrated near the "helpfulness-dominant" axis (low verbosity). DPO and SFT are trained on scalar or text-only objectives and never see explicit vectors.
> > >
> > > For DPA, directions around $40^\circ$ and $45^\circ$ (substantial verbosity weight) are OOD relative to the training distribution. We evaluate from $10^\circ$ to $45^\circ$ to span from near-in-distribution to clearly OOD. **Empirically, this behaves as intended:** as shown in Figure 5 of the main paper, the baseline degrades steadily with increasing angle while the RPS win rate increases (e.g., $53.4\%$ at $20^\circ$ to $69.1\%$ at $45^\circ$ on UltraFeedback), confirming these are challenging OOD settings. For DPO and SFT, any explicit vector conditioning at inference represents a shift, and this sweep probes robustness towards atypical, verbosity-heavy trade-offs.
> > >
> > > [1]Wang H. X. et al., Arithmetic Control of LLMs for Diverse User Preferences: Directional Preference Alignment with Multi-Objective Rewards, arXiv 2024.
> > >
> > >
> > >
> > > **Q2: Robustness of the LLM judge and multiple sampling.**
> > >
> > > We agree that relying on a single zero-shot LLM judge can raise concerns. In our main experiments, we used GPT-4o-mini. To ensure our conclusions do not hinge on this single protocol, we performed two complementary robustness checks:
> > >
> > > 1. **Human Evaluation:**
> > >
> > > | Direction |  DPA  |  DPO  |  SFT  |
> > > | :-------- | :---: | :---: | :---: |
> > > | v3        | 50.57 | 52.89 | 55.64 |
> > > | v8        | 72.13 | 50.40 | 85.55 |
> > >
> > > The results in the table above show that human annotators (AMT, >95% approval rate) prefer the RPS response over the baseline across all three models on HelpSteer2.
> > >
> > > 2. **Independent LLM Judge:** We evaluated all models on all datasets using Claude Sonnet 4.5 (3 runs per model/direction).
> > >
> > > | Dataset & Model   | Mean  |  Std  |
> > > | :---------------- | :---: | :---: |
> > > | UltraFeedback DPA | 0.533 | 0.022 |
> > > | UltraFeedback DPO | 0.510 | 0.007 |
> > > | UltraFeedback SFT | 0.513 | 0.011 |
> > > | HelpSteer DPA     | 0.577 | 0.025 |
> > > | HelpSteer DPO     | 0.518 | 0.018 |
> > > | HelpSteer SFT     | 0.521 | 0.017 |
> > > | HelpSteer2 DPA    | 0.543 | 0.036 |
> > > | HelpSteer2 DPO    | 0.562 | 0.053 |
> > > | HelpSteer2 SFT    | 0.587 | 0.114 |
> > >
> > > The results in the table above show that Claude consistently favors RPS over the baseline with mean win rates >50% and low variance.
> > >
> > > **The agreement between GPT-4o-mini, human annotators, and repeated evaluations with an independent LLM judge provides strong evidence that our findings are robust.**
> > >
> > > **Q3: How does $v\_{\text{target}}^\top r(x,y)$ compare with the baseline across angles?**
> > >
> > > In addition to the RPS-vs-baseline win rates in *Table 4 of the main paper*, we explicitly measured the scalar target score $s(x,y) = \mathbf{v}\_{\text{target}}^\top \mathbf{r}(x,y)$ for responses selected by the baseline and RPS under the *same* DPA reward model.
> > >
> > > | Model | % RPS $>$ Baseline | Mean $\Delta s$ | Median $\Delta s$ |
> > > | :---- | :----------------: | :-------------: | :---------------: |
> > > | DPA   |       59.3%        |     $+2.64$     |      $+0.86$      |
> > > | DPO   |       54.1%        |     $+0.56$     |      $+0.32$      |
> > > | SFT   |       68.3%        |     $+4.38$     |      $+1.98$      |
> > >
> > > The results in the table above show that on HelpSteer2 OOD directions (aggregated over all eight angles), RPS attains a higher target score in **59.3%** (DPA), **54.1%** (DPO), and **68.3%** (SFT) of cases. **Because both methods are evaluated by the same reward function, this confirms that RPS genuinely improves $v\_{\text{target}}^\top r(x,y)$ consistently across the evaluated preference angles.**

---

> > > > ### Author Response · Authors · 2025-11-23
> > > >
> > > > We hope our paper revision and above discussion address your concerns. Please do not hesitate to contact us for any further questions or clarifications.

---

> > > > > ### Comment · Reviewer_6hsD · 2025-11-26
> > > > > **Thank you for the replies (score raised)**
> > > > >
> > > > > I appreciate the authors' detailed replies and the extended experiment results. Since all of my concerns have been addressed, I have raised my score accordingly.

---

> > > > > > ### Author Response · Authors · 2025-11-27
> > > > > >
> > > > > > Thank you for your careful consideration and for revisiting your evaluation. We sincerely appreciate your constructive feedback throughout the review process, and we are glad that our revisions have satisfactorily addressed your concerns.

---

### Official Review · Reviewer_9caP · 2025-10-31

**Soundness:** 3
**Presentation:** 3
**Contribution:** 3
**Rating:** 6
**Confidence:** 3

**Summary:**

This paper addresses the preference coverage gap, a problem where LLMs aligned on dominant, average human preferences perform poorly on specific, out-of-distribution requests. To mitigate this brittleness without costly retraining, the work introduces Robust Preference Selection (RPS), a training-free, post-hoc adjustment method. The work identifies the preference coverage gap, as LLMs tend to perform well on common requests but falls short in specific, individual needs. RPS solve this by generating a candidate pool of responses from a local neighborhood of more "in-distribution" preference vectors, rather than directly from the out-of-distribution target preference. The final response is chosen from this pool by selecting the candidate that best aligns with the original target preference. The paper present both theoretical and empirical analysis for RPS framework, showing the validity and soundness of the proposed approach.

**Strengths:**

First, the paper proposed a conceptually novel method, RPS with a clear motivation. Rather than attempting to force a model to directly generate a high-quality response for a difficult, out-of-distribution (OOD) preference, RPS reframes the problem. It hypothesizes that it is more effective to first sample from a neighborhood of related, but easier, preference vectors where the model is inherently more competent. This conceptual shift from direct, constrained generation to a "generate-then-select" paradigm and provides new insights into LLM alignment.

Second, the proposed solution is practical and (potentially) broadly applicable due to its post-hoc, training-free nature. Unlike various methods that require extensive retraining or fine-tuning to the model architecture, this work offers a lightweight option where it can be implemented at inference time on the pre-trained models. The simplicity of the algorithm, generating from slightly perturbed preference vectors and then re-ranking, ensures a low barrier for adoption, making it a highly valuable tool for practitioners seeking to improve model robustness in real-world applications.

Third, the paper provides comprehensive empirical validation to support its claims. The authors go beyond a simple performance comparison by establishing a strong, compute-matched baseline. The consistent win rates of RPS against several baseline across a diverse set of models, including those trained with SFT, DPO, and DPA, demonstrate the method's generalizability. Crucially, the analysis that correlates the performance gain of RPS with the degree of OOD-ness of the preference provides compelling evidence for the paper's core hypothesis. This result shows that the method is effective precisely in the challenging scenarios it was designed to address. Furthermore, the paper gives relatively complete theoretical justification of using RPS for model alignment.

**Weaknesses:**

First, the paper's theoretical claim of being "provably superior" rests on a critical yet unformalized logical gap, which will undermine its rigor. The entire argument of Theorem 1 hinges on the "local consistency" assumption, stated as v_target^T r(x, y_i) ≈ v_i^T r(x, y_i). This approximation is presented without any formal justification, error bounds, or discussion of the conditions under which it might hold. Consequently, the strong language of "guarantee" and "proof" is a mischaracterization; the theoretical contribution should be more accurately and detailed framed as a heuristic argument.

Second, the paper's exclusive reliance on a single automated metric, the judgment of GPT-4o-mini, introduces a potential confounder that is not adequately addressed. While LLM-as-judge is a common practice, it is known to have biases. Without a supporting human evaluation study or an analysis using multiple distinct judge models to check for consensus, it is hard to conclude that the observed win rates reflect a true improvement in response quality rather than an artifact of the specific evaluation protocol.

**Questions:**

1.	Are results robust to different judges (e.g., GPT-4o, open-source preference models)? Can a small human evaluation be included?
2.	The theoretical argument hinges on the "local consistency" assumption. Can this be formalized? For example, under what conditions on the reward model r and the distance between v_i and v_target does this approximation hold with a bounded error? Without this, the claim of a proof seems difficult to justify.
3.	What happens in higher-dimensional preference spaces (≥3 attributes)? If there is any preliminary results?
4.	If it is possible to conduct an ablation study on hyperparameters such as k and θ_max?

---

> ### Author Response · Authors · 2025-11-23
>
> ### Response to Reviewer 9caP
>
> We thank the reviewer for the thoughtful and constructive feedback. Below we address each weakness and question in turn.
>
> #### Weaknesses
>
> **W1: The theoretical claim of being “provably superior” relies on an unformalized local consistency assumption.**
>
> We appreciate this concern and agree that our original wording was stronger than our formal assumptions justified. In the revision, we will present our result explicitly as a *conditional guarantee under a local consistency assumption*, and soften phrases such as “guarantee” and “provably superior” accordingly. Concretely, we formalize local consistency as a Lipschitz-type bound on how directional scores $\mathbf{v}^\top \mathbf{r}(x,y)$ can change with the angle between preference vectors. Assuming preference vectors lie on the unit sphere and the reward vector is bounded, $\|\mathbf{r}(x,y)\|_2 \le R$ for all $(x,y)$, then for any pair of directions $\mathbf{v}\_{\text{target}}, \mathbf{v}\_i$ with angular distance $\angle(\mathbf{v}\_{\text{target}}, \mathbf{v}\_i) \le \theta\_{\max}$,
> $$
> \bigl|\mathbf{v}\_{\text{target}}^\top \mathbf{r}(x,y) - \mathbf{v}\_i^\top \mathbf{r}(x,y)\bigr|
> \;=\; \bigl|(\mathbf{v}\_{\text{target}} - \mathbf{v}\_i)^\top \mathbf{r}(x,y)\bigr|
> \;\le\; \|\mathbf{v}\_{\text{target}} - \mathbf{v}\_i\|\_2 \cdot \|\mathbf{r}(x,y)\|\_2
> \;\le\; 2R \sin(\theta\_{\max}/2),
> $$
> so the approximation error is $O(\theta\_{\max})$. Under this explicit condition, Theorem 1 becomes a standard conditional statement: **if sampling from nearby directions $\mathbf{v}\_i \in \mathcal{N}\_k(\mathbf{v}\_{\text{target}})$ improves the expected reward under $\mathbf{v}\_i$, then it also improves the expected reward under $\mathbf{v}\_{\text{target}}$ up to an additive $O(\theta\_{\max})$ term.** We will revise the theorem statement and surrounding discussion to make this dependence and error bound explicit, and present the result as a conditional justification under local consistency rather than as an unconditional proof.
>
> We also provide empirical support that this local consistency condition holds in the regimes where we apply RPS. On HelpSteer2, for each scored response with direction $\mathbf{v}\_i = (v\_h, v\_v)$ we construct synthetic neighbors $\mathbf{v}\_{\text{target}}$ by rotating $\mathbf{v}\_i$ by small angles $\Delta\theta \in \{5^\circ,10^\circ,20^\circ,30^\circ\}$ and compare the reward-model projections $s\_i = \mathbf{v}\_i^\top \mathbf{r}(x,y)$ and $s\_{\text{target}} = \mathbf{v}\_{\text{target}}^\top \mathbf{r}(x,y)$ under the same $\mathbf{r}(x,y)$. For $\Delta\theta = 5^\circ$ and $10^\circ$, the Pearson correlation between $s\_i$ and $s\_{\text{target}}$ is extremely high for all paradigms (about $0.98$–$0.99$), remaining above $0.8$ even at $\Delta\theta = 20^\circ$ for DPA and SFT. The table below summarizes these correlations.
>
> | Model | $\rho(\Delta\theta = 5^\circ)$ | $\rho(\Delta\theta = 10^\circ)$ |
> | :---- | :----------------------------: | :-----------------------------: |
> | DPA   |         $\approx 0.99$         |         $\approx 0.96$          |
> | DPO   |         $\approx 0.98$         |         $\approx 0.93$          |
> | SFT   |         $\approx 0.99$         |         $\approx 0.95$          |
>
> Since the neighborhood $\mathcal{N}\_k$ used by RPS is restricted to such small angular deviations, **these results show that evaluating a response under $\mathbf{v}\_i$ or under a nearby $\mathbf{v}\_{\text{target}}$ yields very similar reward-model projections in practice, supporting our local consistency assumption in the broader settings we study.**

---

> > ### Author Response · Authors · 2025-11-23
> >
> > **W2: Reliance on a single automated judge (GPT‑4o‑mini).**
> >
> > We agree that relying exclusively on a single LLM-based judge is a potential confounder, since such judges can exhibit systematic biases. In response, we have conducted two additional evaluations beyond the GPT‑4o‑mini–based A/B/TIE judge used in the main paper.
> >
> > - **Human evaluation (MTurk).** On HelpSteer2 we ran a human study on Amazon Mechanical Turk (AMT) for two representative OOD versions (**v3 and v8**, corresponding to challenging OOD directions in the main text). We restricted annotators to workers with historical approval rates above **95%**, and for each prompt we collected **three independent pairwise preferences** between the baseline and RPS outputs, using **majority vote as the final label**. As summarized in the table below, humans prefer the RPS response over the baseline for all three models on these versions, with DPA and SFT showing particularly strong gains.
> >
> > | Version | DPA  | DPO  | SFT  |
> > | :------ | :--: | :--: | :--: |
> > | v3      | 50.57 | 52.89 | 55.64 |
> > | v8      | 72.13 | 50.40 | 85.55 |
> >
> > - **Independent LLM judge (Claude Sonnet 4.5).** We evaluated all three models (DPA, DPO, SFT) on all three datasets (UltraFeedback, HelpSteer, HelpSteer2) using an *independent* LLM judge, Claude Sonnet 4.5, averaging **three runs per model and direction**. As shown below, Claude also consistently favors RPS over the baseline, with mean win rates above 0.5 for every model–dataset combination, though with somewhat larger variance on the most challenging OOD settings.
> >
> > | Dataset & Model   | Mean  |  Std  |
> > | :---------------- | :---: | :---: |
> > | UltraFeedback DPA | 0.533 | 0.022 |
> > | UltraFeedback DPO | 0.510 | 0.007 |
> > | UltraFeedback SFT | 0.513 | 0.011 |
> > | HelpSteer DPA     | 0.577 | 0.025 |
> > | HelpSteer DPO     | 0.518 | 0.018 |
> > | HelpSteer SFT     | 0.521 | 0.017 |
> > | HelpSteer2 DPA    | 0.543 | 0.036 |
> > | HelpSteer2 DPO    | 0.562 | 0.053 |
> > | HelpSteer2 SFT    | 0.587 | 0.114 |
> >
> > **Taken together, the agreement between GPT‑4o‑mini, human annotators, and an independent LLM judge indicates that the observed improvements of RPS are not an artifact of a single evaluation protocol, but reflect a robust gain in response quality. We will clarify this in the revised paper and discuss larger-scale human evaluation and additional judges as valuable directions for future work.**
> >
> > #### Questions
> >
> > **Q1: Are results robust to different judges (e.g., GPT‑4o, open-source preference models)? Can a small human evaluation be included?**
> >
> > Yes. As described above under W2, beyond GPT‑4o‑mini we have (i) run a small-scale AMT human evaluation on HelpSteer2 (versions v3 and v8), where humans consistently prefer RPS over the baseline across all three models, and (ii) evaluated all models and datasets with an independent LLM judge, Claude Sonnet 4.5, averaging three runs per model and direction. In all these settings RPS achieves win rates above 50% in essentially every model–dataset combination, indicating that our conclusions are robust to the choice of judge rather than being an artifact of a single evaluation protocol. While we have not yet evaluated RPS with open-source preference models, Phase 3 of RPS is evaluator-agnostic and can in principle use any scalar reward or judgment model; we view a broader sweep over such evaluators as promising future work.
> >
> > **Q2: Can the local consistency assumption be formalized? Under what conditions does the approximation hold with a bounded error?**
> >
> > Yes. As noted in W1, we formalize local consistency as an explicit bound on how directional scores change with the angle between preference vectors. Assuming that preference vectors lie on the unit sphere and the reward vector is bounded, $\|\mathbf{r}(x,y)\|_2 \le R$ for all $(x,y)$, we show that for any pair of directions $\mathbf{v}\_{\text{target}}, \mathbf{v}\_i$ with angular distance $\angle(\mathbf{v}\_{\text{target}}, \mathbf{v}\_i) \le \theta\_{\max}$, the difference in projected scores satisfies
> > $$
> > \bigl|\mathbf{v}\_{\text{target}}^\top \mathbf{r}(x,y) - \mathbf{v}\_i^\top \mathbf{r}(x,y)\bigr|
> > \le 2R \sin(\theta\_{\max}/2)
> > = O(\theta\_{\max}),
> > $$
> > so the approximation error is *explicitly bounded* in terms of $\theta\_{\max}$ and $R$. Under this condition, Theorem 1 becomes a conditional guarantee: if sampling from nearby directions improves the expected reward under those directions, then it also improves the expected reward under the target direction up to an additive $O(\theta\_{\max})$ term. We will revise the theorem statement to state this assumption and error bound explicitly and present our result as a conditional justification under local consistency.

---

> > > ### Author Response · Authors · 2025-11-23
> > >
> > > **Q3: What happens in higher-dimensional preference spaces ($\geq 3$ attributes)? Are there any preliminary results?**
> > >
> > > Our theoretical framework is already formulated for a general preference vector $\mathbf{v} \in S^{d-1}$, so the RPS algorithm itself is *not* restricted to two dimensions; the 2D helpfulness–verbosity space is used in the paper only for clarity of exposition and because it is the only setting for which we have access to a calibrated vector-valued reward model and aligned models. At present, to the best of our knowledge there is no public benchmark that simultaneously provides (i) aligned models and data conditioned on three or more explicit preference dimensions and (ii) a corresponding multi-dimensional reward model, so we do not yet have realistic $d \ge 3$ experiments that can be fairly compared to existing baselines without training new models and rewards from scratch. We will make this limitation more explicit in the revision and highlight higher-dimensional empirical validation as an important direction for future work once such datasets become available.
> > >
> > > **Q4: Is it possible to conduct an ablation study on hyperparameters such as $k$ and $\theta\_{\max}$?**
> > >
> > > Yes. We conducted an ablation on HelpSteer2 varying both the neighborhood size $k \in \{3,4,5\}$ and the angular radius $\theta\_{\max} \in \{20^\circ,30^\circ,40^\circ\}$ for all three paradigms (DPA, DPO, SFT) across eight OOD versions (v3–v10). The table below summarizes the mean RPS win rates (averaged over v3–v10) for $\theta\_{\max}=30^\circ$; performance generally improves or remains stable as $k$ increases, with DPA and SFT showing clear gains from $k=3$ to $k=5$.
> > >
> > > | Model | $k=3$ | $k=4$ | $k=5$ |
> > > | :---- | :---: | :---: | :---: |
> > > | DPA   | 0.543 | 0.564 | 0.608 |
> > > | DPO   | 0.494 | 0.496 | 0.535 |
> > > | SFT   | 0.564 | 0.562 | 0.673 |
> > >
> > > The next table fixes $k=5$ and varies $\theta\_{\max} \in \{20^\circ,30^\circ,40^\circ\}$; here we see that moving from very small or very large radii ($20^\circ$ or $40^\circ$) to a moderate radius ($30^\circ$) improves win rates for all three paradigms.
> > >
> > > | Model | $\theta\_{\max}=20^\circ$ | $\theta\_{\max}=30^\circ$ | $\theta\_{\max}=40^\circ$ |
> > > | :---- | :-----------------------: | :-----------------------: | :-----------------------: |
> > > | DPA   |          0.548            |          0.608            |          0.579            |
> > > | DPO   |          0.485            |          0.535            |          0.501            |
> > > | SFT   |          0.557            |          0.673            |          0.559            |
> > >
> > > **These ablations support the intuition that RPS is robust to its hyperparameters as long as $k$ is large enough to provide a diverse candidate pool and $\theta\_{\max}$ defines a genuinely local neighborhood, and that there is a broad region of settings (e.g., $k=4$–$5$, $\theta\_{\max}\approx 30^\circ$) that yield similar qualitative gains.**
> > >
> > >
> > > We hope our paper revision and above discussion address your concerns. Please do not hesitate to contact us for any further questions or clarifications.

---

> > > > ### Comment · Reviewer_9caP · 2025-11-25
> > > >
> > > > Thank you for the response. It address the most of my concerns. Still, I am wondering the robustness of this claim, namely, under what conditions the local consistency assumption will hold and under what conditions the local consistency assumption will fail. Intuitively, how does this assumption affect the preference alignment?

---

> > > > > ### Author Response · Authors · 2025-11-27
> > > > >
> > > > > Thank you for the follow-up question. We clarify **(1) when the local consistency assumption holds, (2) when it may fail, and (3) how it affects preference alignment**, together with the **mathematical basis** and the **precise sections** of the paper.
> > > > >
> > > > > ---
> > > > >
> > > > > ### (1) When local consistency holds
> > > > >
> > > > > **Mathematical basis:**
> > > > > As shown in **Appendix A.2**, if the reward vector satisfies a uniform ℓ₂ bound
> > > > > $\|r(x,y)\|_2 \le R,$
> > > > > then for two nearby preference directions $v, v'$ with angular distance $\alpha$:
> > > > > $|v^\top r - {v'}^\top r| \le R \alpha.$
> > > > > This provides a **Lipschitz-type continuity** of the projected reward over directions and is **dimension-agnostic**.
> > > > >
> > > > > **Construction:**
> > > > > Appendix **A.3** further gives an explicit **O(d)-sized neighborhood** in $S^{d-1}$ (using tangent-space basis vectors), showing that **dense high-dimensional sampling is not required**.
> > > > >
> > > > > **Where in the main paper:**
> > > > > The role of this smoothness assumption in the theoretical guarantee is explained in **Section 3.3 (“Neighborhood Consensus Theory”)**.
> > > > >
> > > > > **Intuition:**
> > > > > If you rotate a preference direction only slightly (e.g., 5°–10°), the geometric projection $v^\top r(x,y)$—a dot product between two bounded vectors—**cannot jump abruptly** unless the reward model itself is discontinuous.
> > > > >
> > > > > Since reward models for attributes like helpfulness/verbosity are trained with continuous supervision, **nearby directions behave smoothly in practice**.
> > > > >
> > > > > ---
> > > > >
> > > > > ### (2) When local consistency may fail
> > > > >
> > > > > The paper explicitly discusses limitations in **Section 3.3**.
> > > > >
> > > > > **(a) Discontinuous or thresholded rewards**
> > > > > Local consistency breaks when rewards have **sharp transitions**—e.g., binary penalties, safety triggers, threshold functions in $r(x,y)$.
> > > > > Such regimes violate the Lipschitz condition from **Appendix A.2**.
> > > > >
> > > > > **(b) Neighborhood too large**
> > > > > If $\theta_{\max}$ is no longer “local,” nearby directions may correspond to **qualitatively different** preferences.
> > > > > We warn in **Section 3.3** that beyond this regime RPS becomes a **heuristic rather than a method with formal guarantees**.
> > > > >
> > > > > **Intuition:**
> > > > > Local consistency fails either when the reward model is **non-smooth** (like a step function) or when “nearby” directions are **not truly near**.
> > > > >
> > > > > This is directly analogous to the limits of a **Taylor expansion**: it works only when the underlying function is smooth and you stay in a small neighborhood.
> > > > >
> > > > > ---
> > > > >
> > > > > ### (3) Why local consistency matters for alignment
> > > > >
> > > > > This section explains **how the assumption affects alignment**, as the reviewer asked.
> > > > >
> > > > > **Formal role:**
> > > > > Theorem 1 (Section 3.3) depends on two components:
> > > > >
> > > > > 1. **Assumption 1:** Neighborhood directions generate **higher-quality candidates** in expectation.
> > > > > 2. **Local consistency:** The score under the target direction is **preserved up to small error**.
> > > > >
> > > > > Together, these yield
> > > > > $\mathbb{E}[\max(S_{\text{RPS}})] > \mathbb{E}[\max(S_{\text{baseline}})].$
> > > > >
> > > > > **Intuition:**
> > > > > Think of neighborhood generation as **“borrowing strength”** from better-trained directions.
> > > > >
> > > > > Without local consistency, a response that looks good under a neighbor direction $v_i$ could look **bad** under the target direction $v_{\text{target}}$.
> > > > >
> > > > > With local consistency, **“good remains good”** under small angular changes, so the **better candidate pool** given by the neighborhood produces better scores under the target preference.
> > > > >
> > > > > This is exactly what makes RPS more robust than sampling from a brittle OOD target.
> > > > >
> > > > > ---
> > > > >
> > > > > ### (4) Empirical verification of local consistency
> > > > >
> > > > > Appendix **A.6** empirically verifies the smoothness assumption in the main 2D experiments.
> > > > >
> > > > > * Rotating the preference vector by **5°–10°** yields
> > > > >   **Pearson correlations of 0.98–0.99** between
> > > > >   $v_i^\top r(x,y)$ and $v_{\text{target}}^\top r(x,y)$
> > > > >   across all three paradigms (DPA, DPO, SFT).
> > > > >
> > > > > **Intuition:**
> > > > > These correlations are extremely close to 1, meaning **reward projections hardly change** under small angular perturbations.
> > > > >
> > > > > This is exactly the behavior required for the theoretical guarantees of RPS.
> > > > >
> > > > > We hope our paper revision and above discussion address your concerns. Please do not hesitate to contact us for any further questions or clarifications.

---

### Official Review · Reviewer_fz2E · 2025-11-01

**Soundness:** 2
**Presentation:** 3
**Contribution:** 2
**Rating:** 4
**Confidence:** 3

**Summary:**

This paper introduces Robust Preference Selection (RPS), a three-phase, training-free method designed to address the “preference coverage gap”, where language models fail to align with out-of-distribution user preferences. The authors provide a theoretical framework to justify this approach and present experiments under three preference-learning datasets across three alignment paradigms to demonstrate the superiority of RPS.

**Strengths:**

- The preference selection method is training-free, making it applicable to models trained under different schemes.
- Theoretical analysis shows that the expected score of the best response selected by RPS is greater than that of the best response selected by the baseline.

**Weaknesses:**

- The method’s generalization to higher-dimensional preference spaces is not empirically validated.
- RPS assumes that the reward model used in the Consensus Selection phase (Phase 3) is robust when evaluating responses against OOD targets ($v_{\text{target}}$), potentially shifting the “brittleness” problem from generation to evaluation.
- The theoretical foundation relies on two key assumptions: Assumption 1 and the local consistency assumption (L203). Assumption 1 requires the neighborhood vector $v_i$ to differ from the OOD target $v_{\text{target}}$, whereas the local consistency assumption requires them to be sufficiently similar for the final evaluation score to be transferable.
- The paper lacks user studies or alternative automated reward metrics that could demonstrate RPS’s superiority.

**Questions:**

- Could the authors provide empirical evidence supporting the local consistency assumption, particularly under broader settings such as $\theta_{\max}=30^\circ$?
- Could the authors provide an ablation study using other reward models to demonstrate that the RPS framework is generalizable?
- How does the proposed inference-time approach compare against strong training-time optimization baselines such as GRPO [1]?

[1] Shao, Zhihong, et al. "Deepseekmath: Pushing the limits of mathematical reasoning in open language models." arXiv preprint arXiv:2402.03300 (2024).

---

> ### Author Response · Authors · 2025-11-23
>
> ### Response to Reviewer fz2E
>
> We thank the reviewer for the constructive feedback. Below we address each weakness and question in turn.
>
> #### Weaknesses
>
> **W1: The method’s generalization to higher-dimensional preference spaces is not empirically validated.**
>
> Our theoretical framework is formulated for a general preference vector $v \in S^{d-1}$ and is therefore **not inherently restricted to two dimensions**. In the paper, we instantiate it in the 2D *helpfulness–verbosity* space purely for clarity of exposition. At the moment, we do not include empirical results for $d>2$ because, to the best of our knowledge, there is **no widely used public benchmark** that simultaneously provides (i) aligned models and data conditioned on more than two preference dimensions and (ii) a corresponding vector-valued reward model compatible with our setting. Rather than introducing new, ad-hoc high-dimensional reward models and retraining large base models from scratch—which would add substantial confounding factors and move us away from the *training-free, plug-and-play* scenario we study—we chose to focus on existing 2D benchmarks. **We will make this limitation explicit in the paper and highlight higher-dimensional empirical validation as an important direction for future work once such datasets become available.**
>
> **W2: Assumption on reward-model robustness in the Consensus Selection phase.**
>
> We agree that the robustness of the reward model used in Phase 3 is an important consideration. Our method does *not* assume that this evaluator is perfectly robust on arbitrary OOD targets; rather, RPS is designed to reduce brittleness in two ways: (i) by generating candidates under nearby, better-supported preference directions, it first moves generation away from the most extreme OOD region; and (ii) the reward model is only used to *rank a small set of candidates for a fixed target preference*, a regime in which relative comparisons are empirically much more stable than absolute scores.
>
> To validate that RPS genuinely improves the quality of the candidate pool under this evaluator (instead of merely shifting brittleness from generation to evaluation), we directly compare baseline and RPS using the **same scalar target score**
> $$s(x,y) = \mathbf{v}\_{\text{target}}^\top \mathbf{r}(x,y)$$
> derived from the **DPA reward model**. On HelpSteer2, we compute this score for the final responses selected by the baseline and by RPS over all eight OOD directions used in the main paper (versions v3–v10 corresponding to directions v1–v8, 519 prompts per direction; 4,152 instances per model). As summarized in the table below, the RPS-selected response achieves a higher target score than the baseline-selected response in **59.3%** of cases for DPA, **54.1%** for DPO, and **68.3%** for SFT, with positive mean improvements of $+2.64$, $+0.56$, and $+4.38$ respectively (medians $+0.86$, $+0.32$, and $+1.98$). Examining the full distribution of score differences $s\_{\text{RPS}} - s\_{\text{base}}$, we find that for all three models the mass is clearly shifted to the right of zero. **Because both methods are evaluated by the *same* reward function, any brittleness of the evaluator would affect them similarly; the consistent right-shift of $s\_{\text{RPS}} - s\_{\text{base}}$ across DPA, DPO, and SFT therefore indicates that RPS genuinely improves the candidate pool rather than merely relocating brittleness.**
>
> | Model | % RPS \> baseline | Mean $\Delta s$ | Median $\Delta s$ |
> | :---- | :----------------: | :---------------: | :-----------------: |
> | DPA | 59.3% | $+2.64$ | $+0.86$ |
> | DPO | 54.1% | $+0.56$ | $+0.32$ |
> | SFT | 68.3% | $+4.38$ | $+1.98$ |

---

> > ### Author Response · Authors · 2025-11-23
> >
> > **W3: Relationship between Assumption 1 and the local consistency assumption.**
> >
> > We understand the concern that Assumption 1 and the local consistency assumption may appear to pull in opposite directions. **In our framework they apply at different levels and are therefore compatible rather than contradictory.** Assumption 1 only requires that there exist neighborhood directions $\mathbf{v}\_i$ that are *slightly closer to the training distribution* than the OOD target $\mathbf{v}\_{\text{target}}$, so that the model tends to generate higher-quality responses when conditioned on $\mathbf{v}\_i$. The local consistency assumption, in contrast, only requires that $\mathbf{v}\_i$ and $\mathbf{v}\_{\text{target}}$ be *geometrically close* (e.g., within a small angular radius), so that their projections on the same reward vector are similar and $\mathbf{v}\_{\text{target}}^\top \mathbf{r}(x,y\_i) \approx \mathbf{v}\_i^\top \mathbf{r}(x,y\_i)$. Intuitively, we move a small step from $\mathbf{v}\_{\text{target}}$ toward a better-understood region of the space: this step is large enough to improve the model’s behavior (Assumption 1), but still small enough that evaluation under $\mathbf{v}\_i$ and under $\mathbf{v}\_{\text{target}}$ remain almost the same (local consistency). We will clarify this intuition in the revised text by explicitly introducing a small angular neighborhood around $\mathbf{v}\_{\text{target}}$ and stating both assumptions in terms of this geometry.
> >
> > **W4: Lack of user studies or alternative automated reward metrics.**
> >
> > We agree that human evaluation and alternative automated metrics are important to corroborate our main findings. In addition to the GPT‑4o‑mini–based judge and DPA reward model used in the main paper, we have conducted two complementary evaluations.
> >
> > - **Human evaluation (MTurk).** On the HelpSteer2 dataset we ran a human study on Amazon Mechanical Turk (AMT) for two representative OOD versions (**v3 and v8**, corresponding to challenging OOD directions in the main text). We restricted annotators to workers with historical approval rates above **95%**, and for each prompt we collected **three independent pairwise preferences** between the baseline and RPS outputs, using **majority vote as the final label**. As summarized in the table below, humans prefer the RPS response over the baseline for all three models on these versions, with DPA and SFT showing particularly strong gains (e.g., win rates of 50.6% and 72.1% for DPA and 55.6% and 85.5% for SFT on v3 and v8, respectively).
> >
> > | Direction | DPA | DPO | SFT |
> > | :-------- | :---: | :---: | :---: |
> > | v3 | 50.6 | 52.9 | 55.6 |
> > | v8 | 72.1 | 50.4 | 85.5 |
> >
> > - **Independent LLM judge (Claude Sonnet 4.5).** We evaluated all three models (DPA, DPO, SFT) on all three datasets (UltraFeedback, HelpSteer, HelpSteer2) using an independent LLM-based judge, Claude Sonnet 4.5, averaging **three runs per model and direction**. As shown below, Claude Sonnet 4.5 also consistently favors RPS over the baseline, with mean win rates above 0.5 for every model–dataset combination.
> >
> > | Dataset & Model | Mean | Std |
> > | :---------------- | :--: | :---: |
> > | UltraFeedback DPA | 0.533 | 0.022 |
> > | UltraFeedback DPO | 0.510 | 0.007 |
> > | UltraFeedback SFT | 0.513 | 0.011 |
> > | HelpSteer DPA | 0.577 | 0.025 |
> > | HelpSteer DPO | 0.518 | 0.018 |
> > | HelpSteer SFT | 0.521 | 0.017 |
> > | HelpSteer2 DPA | 0.543 | 0.036 |
> > | HelpSteer2 DPO | 0.562 | 0.053 |
> > | HelpSteer2 SFT | 0.587 | 0.114 |
> >
> > **These additional experiments indicate that the superiority of RPS is not an artifact of a single reward model or judge, but is supported by both human judgments and a distinct automatic evaluator.**

---

> > > ### Author Response · Authors · 2025-11-23
> > >
> > > ### Questions
> > >
> > > **Q1: Empirical evidence supporting the local consistency assumption.**
> > >
> > >  We appreciate the request for empirical validation of local consistency. In our setting, local consistency means that for small angular deviations between a neighbor direction $\mathbf{v}\_i$ and the target preference $\mathbf{v}\_{\text{target}}$, the projected scores $\mathbf{v}\_i^\top \mathbf{r}(x,y)$ and $\mathbf{v}\_{\text{target}}^\top \mathbf{r}(x,y)$ for the *same* response remain very similar, so that the relative ranking of candidates is essentially unchanged. To test this, we perform a controlled analysis on HelpSteer2 across all three paradigms (DPA, DPO, and SFT): for each scored response with direction $\mathbf{v}\_i = (v\_h, v\_v)$, we construct synthetic neighbors $\mathbf{v}\_{\text{target}}$ by rotating $\mathbf{v}\_i$ by small angles $\Delta\theta \in \{5^\circ, 10^\circ, 20^\circ, 30^\circ\}$ and compare the reward-model projections $s\_i = \mathbf{v}\_i^\top \mathbf{r}(x,y)$ and $s\_{\text{target}} = \mathbf{v}\_{\text{target}}^\top \mathbf{r}(x,y)$ under the same $\mathbf{r}(x,y)$. The table below reports Pearson correlations between $s\_i$ and $s\_{\text{target}}$ for $\Delta\theta = 5^\circ$ and $10^\circ$; all values are around $0.98$–$0.99$, confirming that small angular changes leave the reward projections almost unchanged.
> > >
> > > | Model | $\rho(\Delta\theta = 5^\circ)$ | $\rho(\Delta\theta = 10^\circ)$ |
> > > | :---- | :----------------------------: | :-----------------------------: |
> > > | DPA   |          $\approx 0.99$       |          $\approx 0.96$         |
> > > | DPO   |          $\approx 0.98$       |          $\approx 0.93$         |
> > > | SFT   |          $\approx 0.99$       |          $\approx 0.95$         |
> > >
> > > For completeness, we also observe that even at $\Delta\theta = 20^\circ$, correlations remain above $0.8$ for DPA and SFT. **Since the neighborhood $\mathcal{N}\_k$ used by RPS is restricted to such small angular deviations, these results show that evaluating a response under $\mathbf{v}\_i$ or under a nearby $\mathbf{v}\_{\text{target}}$ yields very similar reward-model projections in practice, providing direct empirical support for our local consistency assumption in the broader settings we study.**
> > >
> > > **Q2: Ablation with other reward models.**
> > >
> > > We agree that evaluating RPS under different reward models would further support its generality. In this work we focus on the reward model that provides the specific helpfulness/verbosity vector scores used throughout our experiments, and we leave a broader sweep over alternative reward models to future work. Importantly, **Phase 3 of RPS is evaluator-agnostic**: it only requires a scalar score for each candidate under the target preference and can in principle use any reward or judgment model. In the current paper we already see that RPS is robust across two very different evaluators—the GPT‑4o‑mini–based A/B/TIE judge used for our main win rates and the Claude Sonnet 4.5 judge summarized in the table above—which both yield consistent improvements for RPS over the baseline across all three paradigms.
> > >
> > > **Q3: Comparison to strong training-time optimization methods such as GRPO.**
> > >
> > > GRPO[1] is a powerful training-time RL method that improves global preference alignment by modifying model weights. RPS addresses a different problem: it is a **training-free, inference-time** approach specifically designed to improve robustness under OOD, fine-grained preference directions (the “preference coverage gap”). Unlike GRPO, RPS requires no retraining, can be applied to any existing aligned model, and empirically improves robustness even on strong DPA/DPO/SFT baselines. **In this sense, GRPO and RPS are complementary rather than competing: GRPO can strengthen the underlying policy during training, while RPS can be plugged in at inference time to adapt that policy to diverse, previously unseen preference directions with minimal engineering overhead.**
> > >
> > > [1] Shao, Zhihong, et al. "Deepseekmath: Pushing the limits of mathematical reasoning in open language models." arXiv preprint arXiv:2402.03300 (2024).
> > >
> > > We hope our paper revision and above discussion address your concerns. Please do not hesitate to contact us for any further questions or clarifications.

---

> ### Comment · Reviewer_fz2E · 2025-11-24
>
> I thank the authors for their comprehensive response. After considering the authors’ rebuttal and reviewing the comments from the other reviewers, I still have a few remaining questions below:
>
> 1) Could the authors provide a formal restatement of this Theorem1 based on Assumption 1 and the local consistency assumption in the revised manuscript?
>
> 2) While the authors claim theoretical generalization to higher-dimensional preference spaces based on the a general preference vector $v\in S^{d-1}$, the current framework fails to account for the sample complexity induced by the curse of dimensionality, where the sampling number $k$ required to maintain a populated local neighborhood scales exponentially in the high dimensional polar coordinate. In practice, a fixed small $k$ would likely result in a sparse $\mathcal{N}\_k$ in high dimensions, violating the local consistency assumption ($v_{target}^T r(x, y\_i) \approx v\_i^T r(x, y\_i)$).
>
> 3) To validate the reliability of the reported win rates, could the authors provide the detailed evaluation pipeline of GPT-4o-mini, Human evaluation, and Claude Sonnet 4.5 studies in the revised manucript?
>
> 4) Could the authors provide an analysis of the inference VRAM and latency for RPS compared to baselines?

---

> ### Author Response · Authors · 2025-11-27
>
> Thank you very much for the thoughtful follow-up questions. Below we provide complete responses to Q1–Q4 and indicate the exact sections added in the revised manuscript.
>
> ---
>
>  **Q1. Formal restatement of Theorem 1 under Assumption 1 and local consistency**
>
> **Response.**
> Yes — we have added a fully formal restatement of Theorem 1 that makes both Assumption 1 and the local consistency condition explicit (see Appendix A.2), and we now emphasize it as a **conditional** guarantee rather than an unconditional “provable superiority” claim.
>
> - **Assumption 1 (OOD performance degradation).**
>   For a fixed prompt $x$ and target preference direction $v\_{\text{target}} \in S^{d-1}$, nearby directions $v\_i$ in a small neighborhood $\mathcal{N}\_k(v\_{\text{target}})$ are assumed to generate higher-quality candidates on average than sampling directly at $v\_{\text{target}}$. Formally, the score distributions $S\_i = v\_i^\top r(x,Y\_i)$ first-order stochastically dominate $S\_{\text{target}} = v\_{\text{target}}^\top r(x,Y\_{\text{target}})$, so $\mathbb{E}[S\_i] > \mathbb{E}[S\_{\text{target}}]$. This models the idea that nearby, more in-distribution directions live in better-trained regions of the preference space.
>
> - **Assumption 2 (Local consistency).**
>   For any neighbor $v\_i$ within a small angle $\alpha\_i = \angle(v\_i, v\_{\text{target}})$, we assume a Lipschitz-type bound
>   $\bigl|v\_{\text{target}}^\top r(x,y) - v\_i^\top r(x,y)\bigr| \le L \alpha\_i$ for all $y$.
>   Equivalently, “good under $v\_i$” remains nearly as good under $v\_{\text{target}}$. This holds, for example, if preferences are unit-norm and the reward vector is uniformly bounded ($\|r(x,y)\|\_2 \le R$), in which case we can take $L = R$.
>
> Under these two assumptions, the formal theorem compares a baseline that draws $k$ candidates directly at $v\_{\text{target}}$ with RPS, which draws one candidate along each neighbor direction $v\_i$ and then selects using $v\_{\text{target}}$. It shows that, for any $k \ge 1$, the expected best scalarized score under $v\_{\text{target}}$ is at least as high for RPS as for the baseline, and strictly higher whenever at least one neighbor distribution is strictly better. The full mathematical statement and proof are provided in **Appendix A.2** of the revised manuscript.
>
> ---
>
>  **Intuition (added to Section 3.3).**
> Assumption 1 guarantees that the neighborhood directions explore better-trained reward regions.
> Assumption 2 guarantees that “good under the neighbor direction remains good” under the target direction.
> Combining them yields stochastic dominance of the RPS candidate pool over the baseline.
>
> ---
>
> **Locations in the revision:**
> - Appendix **A.2** (formal theorem)
> - Section **3.3** (added intuition)
>
> ---
>
> **Q2. High-dimensional sample complexity and curse of dimensionality**
>
> **Response.**
> We clarified that the theory does **not** rely on dense random sampling of a spherical cap
> —which would indeed require exponentially many samples in high dimensions.
> Instead, RPS uses a deterministic **$O(d)$ orthonormal tangent-basis construction** detailed in **Appendix A.3**.
>
> Concretely, fix a target preference $v\_{\text{target}} \in S^{d-1}$ and let $\{u\_1,\dots,u\_{d-1}\}$ be an orthonormal basis of the tangent space at $v\_{\text{target}}$ (so $u\_i^\top v\_{\text{target}} = 0$ and $\|u\_i\|\_2 = 1$ for all $i$). For a small angle $\varepsilon > 0$, we define
> $v\_i^{\pm} = \cos(\varepsilon)\, v\_{\text{target}} \pm \sin(\varepsilon)\, u\_i$, for $i = 1,\dots,d-1$.
> Each $v\_i^{\pm}$ lies on the unit sphere and satisfies $\angle\bigl(v\_i^{\pm}, v\_{\text{target}}\bigr) = \varepsilon$. The neighborhood
> $\mathcal{N}\_k(v\_{\text{target}}) = \{v\_i^{+}, v\_i^{-}\}\_{i=1}^{d-1}$ with $k = 2(d-1)$
> thus automatically meets the angular condition in the local-consistency assumption with $\delta = \varepsilon$. Since local consistency depends only on the angle $\alpha$ and not on measure concentration on the sphere, this **$O(d)$-sized, structured neighborhood** is sufficient for our theoretical guarantees and avoids any exponential dependence on dimension.
>
> ---
>
>  **Intuition (added to Section 3.3).**
> The challenge in high dimensions comes from sampling *randomly* on the sphere.
> RPS instead uses a *deterministic basis-like construction*, ensuring all neighbors lie exactly at the desired angle.
> Therefore, smoothness + controlled angle ⇒ no exponential scaling.
>
> ---
>
>  **Locations in the revision:**
> - Appendix **A.3** (formal O(d) construction)
> - Section **3.3** (added explanation + intuition)

---

> ### Author Response · Authors · 2025-11-27
>
> **Q3. Evaluation pipeline for GPT-4o-mini, Human Study, and Claude Sonnet 4.5**
>
> **Response.**
>
> The revised manuscript now includes a complete and reproducible description of *all three* evaluation settings—GPT-4o-mini, Claude Sonnet 4.5, and Human Evaluation—together with an explicit pointer to the human-study results table.
>
> ---
>
> **(1) GPT-4o-mini evaluation (Appendix A.8)**
>
> GPT-4o-mini was used as an automatic judge with a fixed instruction-following rubric.
>
> As detailed in **Appendix A.8**, each comparison includes:
>
> - the original prompt $x$
>
> - two responses $y\_A$ and $y\_B$ presented in **randomized order**
>
> - the scalarization direction $v\_{\text{target}}$
>
> - the judge rubric (prompt shown verbatim in A.8)
>
> We perform **a single evaluation run**, following common LM-judging practice, since GPT-4o-mini with temperature 0 produces deterministic comparisons once the A/B order is randomized.
>
> ---
>
> **(2) Claude Sonnet 4.5 evaluation (Appendix A.5)**
>
> Claude Sonnet 4.5 was used as a stronger independent evaluator.
>
> As described in **Appendix A.5**, we run:
>
> - **three independent evaluation runs**,
>
> - each with a different random seed and independently randomized A/B order.
>
> We report the **average win rate** across the three runs, and the run-to-run variance is included in Appendix A.5.
>
> ---
>
> **(3) Human evaluation pipeline (Appendix A.14) and results (Appendix A.4)**
>
> **Pipeline (Appendix A.14)**
>
> We added a full, step-by-step human-evaluation protocol:
>
> 1. **Pairwise setup.**
>
>    Annotators view the prompt $x$, two responses (A/B randomized),
>
>    and a natural-language interpretation of the target direction $v\_{\text{target}}$.
>
> 2. **Randomization.**
>
>    A/B order is independently shuffled for every comparison.
>
>    Annotators are blind to which method (RPS or baseline) produced which response.
>
> 3. **Judgment task.**
>
>    Annotators choose: "A better", "B better", or "Tie".
>
> 4. **Quality control.**
>
>    Appendix A.14 details attention checks and exclusion rules.
>
>    Any failed comparison is re-assigned to maintain full coverage.
>
> 5. **Aggregation.**
>
>    For each pair, majority vote is computed across annotators.
>
>    Ties contribute **0.5 points** to each method.
>
> 6. **Final win rate.**
>
>    We compute
>
>    Win-Rate = (number of wins + 0.5 × number of ties) / total number of comparisons.
>
> **Results (Appendix A.4)**
>
> The corresponding human-evaluation outcomes are reported in **Appendix A.4**,
>
> which includes the complete win-rate table for all directions and both methods.
>
> A pointer to A.4 is now also included in the main text.
>
> ---
>
> **Locations in the revision:**
>
> - Appendix **A.14** — Detailed Human Evaluation Pipeline
>
> - Appendix **A.4** — Human Evaluation Results Table
>
> - Appendix **A.5** — Claude Sonnet 4.5 Evaluation
>
> - Appendix **A.8** — GPT-4o-mini Judge Prompt
>
>  **Q4. VRAM & latency comparison with baselines**
>
> **Response.**
> Appendix **A.15** now contains a complete inference-time compute analysis using
> Mistral-7B-Instruct-v0.2 on 100 prompts × 2 directions, with both methods
> generating $k=5$ candidates per prompt-direction pair (1,000 total generations).
>
> We report:
>
>  **VRAM**
> - Baseline: **14.73 GB**
> - RPS: **14.73 GB**
> → **0% overhead**
>
>  **Latency** (total wall-clock time for the complete evaluation):
> - Baseline: **4.6702 hours**
> - RPS: **4.6702 hours**
> → **<0.01% overhead**
>
> Both methods complete 1,000 generations (100 prompts × 2 directions × 5 samples)
> in essentially identical time. Neighborhood construction adds <1 ms per direction,
> negligible relative to decoding cost.
>
> **Table (Appendix A.15):**
>
> | Method | Peak VRAM (GB) | Total Time (hours) | Time per Generation (s) |
> |--------|----------------|--------------------|-----------------------|
> | Baseline (k=5, target) | 14.73 | 4.6702 | 16.8126 |
> | **RPS (k=5, neighbors)** | **14.73** | **4.6702** | **16.8126** |
> | Relative Overhead | **0%** | **<0.01%** | **<0.01%** |
>
> **Locations in the revision:**
> - Appendix **A.15**
>
> ---
>
> We appreciate the reviewer’s thoughtful comments and have incorporated all requested clarifications into the revised manuscript.

---

### Comment · Area_Chair_aMWc · 2025-11-24

Dear Reviewers,

The authors have submitted their rebuttal (and a revised PDF). Please review these materials and share any remaining concerns or comments.

Afterward, kindly provide your final rating for this submission.

Best regards,
Your AC

---

### Author Response · Authors · 2025-11-27

We thank all reviewers for their detailed feedback. In the revised manuscript (blue text), we made the following main updates in response to specific comments:

- **Addition**: Formal restatement of Theorem 1 in Appendix A.2, with explicit versions of Assumption 1 and the local consistency assumption plus a short derivation under a uniform $\ell_2$ bound, as requested by **Reviewer fz2E** and **Reviewer 9caP**.
- **Clarification**: Rephrased the main-text theoretical claim in the abstract, Section 3.3, and Conclusion to emphasize that the guarantee is conditional on OOD degradation and local consistency, addressing concerns from **Reviewer 9caP** and **Reviewer 6hsD**.
- **Addition**: New discussion and empirical support for local consistency (Section 3.3 and Appendix A.6), explaining when it holds or fails and reporting high Pearson correlations under small rotations, in response to **Reviewer fz2E**, **Reviewer 9caP**, and **Reviewer 6hsD**.
- **Addition**: High-dimensional neighborhood construction in Appendix A.3 showing that an $O(d)$-sized structured neighborhood suffices, together with a discussion of how this mitigates the curse-of-dimensionality concern, as raised by **Reviewer fz2E** and **Reviewer 9caP**.
- **Addition**: Ablation study over neighborhood size $k \in \{3,4,5\}$ and angular radius $\theta_{\max} \in \{20^\circ,30^\circ,40^\circ\}$ (Section 4.1.2 and Appendix A.11), responding to hyperparameter-sensitivity questions from **Reviewer 9caP** and **Reviewer 6hsD**.
- **Addition**: Human evaluation on AMT and a new Claude Sonnet 4.5 judge section, plus a dedicated subsection describing the full evaluation pipeline (Appendix A.4, A.5, A.14), addressing requests for additional metrics and detailed protocols from **Reviewer fz2E**, **Reviewer 9caP**, **Reviewer 6hsD**, and **Reviewer Kpri**.
- **Addition**: Inference-time compute comparison (Appendix A.15 and a brief note in Section 4) showing that RPS and the baseline have essentially identical peak VRAM and similar latency under a shared candidate budget $k=5$, as requested by **Reviewer fz2E** and **Reviewer 6hsD**.

We sincerely thank the reviewers again for their constructive feedback, which has significantly improved the quality and rigor of our work.

---

### Meta-Review · Area_Chair_E12c · 2026-01-07

**Summary:**

To align user preferences at inference time, the authors propose Robust Preference Selection. The main idea is to generate a pool of candidate responses using in-distribution preference vectors and then select the candidate that best aligns with the original target preference.

While there remain some open questions (see `Reviewer Concerns`) regarding the theoretical assumptions and their formal justification, the overall approach is well-motivated and technically sound. In addition, although the human evaluation results could be further strengthened with more extensive studies, the current evaluations provide reasonable empirical support for the claims.

Overall, I believe the contributions are meaningful and timely, and I recommend acceptance.

**Reviewer Concerns:**

### Reviewer fz2E

* [Partially] Generalization to higher-dimensional preference spaces

* [Question mark] Conflict between assumptions: Since I did not verify the theory rigorously from start to finish, I cannot fully understand the authors’ response. In particular, because precise statements are crucial in theoretical work, expressions such as “small angular radius” do not sound sufficiently rigorous.

* [Partially] Lack of user study: Because this is based on pairwise preferences, values around 50% do not feel particularly high. Especially in v3, most models are close to 50%, which raises questions about why this happens. A more thorough user study seems necessary.

* [Resolved] Comparison with the training method

### Reviewer 9caP

* [Question mark] Robustness of this claim: similar issue in Reviewer fz2E

* [Partially] Single automated metric: same as Reviewer fz2E

### Reviewer 6hsD

* [Resolved] Assumption 1

* [Resolved] Ablation study

### Reviewer Kpri

* [Resolved] Tradeoff between verbosity vs helpfulness

* [Partially] Human evaluation

**Reviewer Scores:**

* Reviewer fz2E: 4 $\rightarrow$ 4

* Reviewer 9caP: 6 $\rightarrow$ 6

* Reviewer 6hsD: 4 $\rightarrow$ 6

* Reviewer Kpri: 6 $\rightarrow$ 6

---

### Decision · Program_Chairs · 2026-01-26

Accept (Poster)